# Improving CLIP Training with Language Rewrites

**Lijie Fan**[1,2,*]    **Dilip Krishnan**[1]    **Phillip Isola**[2]    **Dina Katabi**[2]    **Yonglong Tian**[1,*]

[1]Google Research,    [2]MIT CSAIL,    [*]equal contribution

## Abstract

Contrastive Language-Image Pre-training (CLIP) stands as one of the most effective and scalable methods for training transferable vision models using paired image and text data. CLIP models are trained using contrastive loss, which typically relies on data augmentations to prevent overfitting and shortcuts. However, in the CLIP training paradigm, data augmentations are exclusively applied to image inputs, while language inputs remain unchanged throughout the entire training process, limiting the exposure of diverse texts to the same image. In this paper, we introduce Language augmented CLIP (LaCLIP), a simple yet highly effective approach to enhance CLIP training through language rewrites. Leveraging the in-context learning capability of large language models, we rewrite the text descriptions associated with each image. These rewritten texts exhibit diversity in sentence structure and vocabulary while preserving the original key concepts and meanings. During training, LaCLIP randomly selects either the original texts or the rewritten versions as text augmentations for each image. Extensive experiments on CC3M, CC12M, RedCaps and LAION-400M datasets show that CLIP pre-training with language rewrites significantly improves the transfer performance without computation or memory overhead during training. Specifically for ImageNet zero-shot accuracy, LaCLIP outperforms CLIP by 8.2% on CC12M and 2.4% on LAION-400M. Code is available at https://github.com/LijieFan/LaCLIP.

## 1   Introduction

Pre-trained vision-language multi-modal encoders, exemplified by CLIP [44], have proven to be extremely useful in learning transferable features from paired image and text data. CLIP's training process can leverage two scalable paradigms: data and compute. Firstly, the availability of large-scale vision-language paired data [49, 48] enables effective training at a substantial scale. Secondly, CLIP's utilization of language and image co-embeddings grants it favorable scaling properties with respect to compute resources [27]. Consequently, CLIP embeddings consistently outperform other visual pre-training approaches such as SimCLR [8] or MAE [22] across various downstream tasks [27]. Follow-up methods that build upon language-image pre-training, such as SLIP [41] and FLIP [35], exhibit similar advantageous scaling performance.

The CLIP framework is built upon contrastive learning, which typically relies on data augmentations to prevent overfitting and the learning of ineffective shortcuts [8, 47]. However, in the CLIP training process, this beneficial feature is applied exclusively to image inputs, which undergo augmentations in every epoch. In contrast, text inputs are neglected and remain unchanged throughout training, lacking any form of augmentation. This input asymmetry leads to scenarios where the same text is consistently paired with slightly augmented images, while the augmented version of the same image is always paired with the exact same words. Such asymmetry presents two issues. Firstly, the image encoders receive limited supervision from the language side since the same image is consistently paired with the same words. Consequently, the language aspect provides less guidance to the image

encoders. Secondly, the text encoders repeatedly encounter the exact same texts in each epoch, which increases the risk of text overfitting and significantly impacts zero-shot transferability.

Hence, it becomes crucial to incorporate a suitable augmentation strategy for the text inputs. However, existing language augmentation methods are not sufficiently effective. Most previous approaches [58] focus on word-level treatments like replacement or masking, which have limited impact on enriching text structures and are comparatively weaker than image augmentations. Currently, there is a lack of language rewriting strategies that can effectively augment sentences while preserving key concepts and meanings. Such strategies are urgently needed for CLIP training to ensure optimal performance.

Meanwhile, alongside the development of CLIP, the field has witnessed significant advancements in large language models (LLMs) like GPT models [5, 44, 45] and LaMDA [3]. These LLMs have seen tremendous growth in terms of data, computational resources, and performance. Instruction fine-tuned models such as ChatGPT [2] and Bard [1] have also emerged, incorporating fine-tuning through supervised and reinforcement learning. These models have exhibited exceptional performance surpassing human capabilities across a wide range of natural language tasks. Motivated by these advancements, we naturally explored the potential of leveraging LLMs to effectively generate diverse rewritten versions of a given text. While a straightforward approach would involve directly employing instruction fine-tuned models like ChatGPT or Bard, their closed-source nature renders it infeasible to use them for rewriting the hundreds of millions of image descriptions in the datasets. Fortunately, open-sourced LLMs such as LLaMA [56], despite lacking fine-tuning with instructions, possess excellent In-Context Learning (ICL) capabilities, enabling predictions with a limited context. By thoughtfully designing contextual examples, LLaMA can generate diverse and rich text rewrites for the entire dataset.

Building upon this foundation, we propose Language augmented CLIP (LaCLIP), a straightforward yet highly effective approach for enhancing CLIP model performance by harnessing the power of LLMs. Our method leverages ICL using LLaMA to generate diverse variants of each caption within the text-image pairs of a given pre-training dataset. To facilitate ICL prompts, we have devised multiple strategies to generate a small set of meta-input-output caption pairs. These strategies involve utilizing ChatBots, human rewriters, or existing image captioning datasets. Once we have acquired the meta-input-output pairs, we employ them as examples to prompt LLaMA, enabling the rewriting of millions of texts within the entire dataset. Unlike existing strategies for text rewriting [58, 50], which tend to preserve the sentence structure, LLMs exhibit the remarkable ability to generate language rewrites with greater richness and diversity. This is attributed to their emergent properties and extensive training data. Following the caption rewriting process conducted by LLaMA ICL, each image is now accompanied by a collection of diverse captions resulting from the rewriting process. Utilizing these rewritten texts, we proceed to train CLIP models with augmentation also on the text side. The text augmentation could be performed by randomly selecting one out of the many captions associated with each image.

Extensive experiments on various pretraining datasets at different scales demonstrate our proposed LaCLIP could significantly improve the transferability of CLIP. For instance, on the LAION-400M dataset [49], we observe a notable improvement over CLIP in the zero-shot performance on ImageNet, increasing from 62.0% to 64.4%. We firmly believe that this strategy presents a simple, scalable approach that contributes to the array of training strategies available for training image embeddings.

## 2 Related Works

**Vision-Language models.** There are a number of earlier works demonstrating the effectiveness of learning visual representations from the supervision of corresponding text [25, 31, 14, 62]. Contrastive Language-Image Pretraining (CLIP) [44] has attracted significant attention due to its superior representation learning and zero-shot transfer ability. This performance is achieved through contrastive learning on top of image and text features. Another related approach is ALIGN [24], which achieves similar performance with larger and noisier datasets. There have been numerous follow-up works that attempt to improve the efficacy and data efficiency of CLIP training. SLIP [41] and De-CLIP [34] proposes to improve the performance by incorporating self-supervised training techniques. FILIP [60] proposes to leverage cross-modal fine-grained alignment between image patches and text words. CoCa [61] introduces an additional decoder and captioning loss. LIT [63] proposes to boost zero-shot transferring performance by fine-tuning the text encoders. BLIP series [33, 32] include

additional captioners and incorporates iterative image captioning within the training pipeline, which intricately link the generated captions with the image content. However, most of these follow-up works introduce additional training inputs and losses, which can have a negative impact on training efficiency and memory consumption.

**Text augmentation and Large Language Models.** The success of natural language processing (NLP) tasks is strongly dependent on the quality and quantity of available data. Deep neural networks benefit from more data [40, 58]. Common practices in data augmentation include synonym replacement [58], random masking [30], and back translation [50]. The advent of self-supervised large language models like BERT [16] and GPT series [45, 46] has been a game-changer, as they do not require labeled data and can scale up to web-level data to achieve superior transfer ability. Recently, even larger foundation models with billion-level parameters have emerged, revolutionizing the NLP community. Models like the 175B GPT-3 [5] and 540B PaLM [10] achieve superior performance on various NLP tasks. GPT-3 also demonstrates the few-shot in-context learning ability of large language models [5]. Open-sourced LLaMA [56] also achieve comparable performances on various benchmarks. In addition to them, ChatGPT and Bard are chatbots trained with reinforcement learning human feedback (RLHF), and have achieved human-comparable performances on various language understanding tasks [6].

# 3 Improving CLIP with Language Rewrites

This section outlines the core design of our LaCLIP framework, highlighting the key components and strategies involved. We provide a comprehensive description of our approach, including the generation of a small set of meta-input-output text pairs from diverse sources, the process of rewriting image descriptions across the entire dataset using LLM ICL, and the enhanced CLIP training strategies incorporating these rewritten descriptions.

## 3.1 Preliminary

**CLIP.** The Contrastive Language-Image Pretraining (CLIP) method has proven to be highly effective to train vision models using language supervision. In this framework, a large batch of $N$ paired images and text $\{x_I, x_T\}$ are sampled from the training dataset during each training step. The images are pre-processed using data augmentations, and image and text features are extracted using dedicated encoders and normalization functions $f_I$ and $f_T$. The image text features are used to compute the InfoNCE loss, where the paired images and text form the positive pairs and the unpaired ones are treated as negative samples. The training loss can be formulated as follows:

$$L_I = -\sum_{i=1}^{N} \log \frac{\exp\left(\text{sim}(f_I(\text{aug}_I(x_I^i)), f_T(x_T^i))/\tau\right)}{\sum_{k=1}^{N} \exp\left(\text{sim}(f_I(\text{aug}_I(x_I^i)), f_T(x_T^k))/\tau\right)}, \tag{1}$$

In this scenario, $(x_I^i, x_T^i)$ is the $i^{th}$ image-text pair, and $\text{aug}_I()$ denotes the image augmentation functions. The $\text{sim}(\cdot, \cdot)$ function measures distance using the dot product, while the temperature $\tau$ is a learnable parameter that scales the logits. To simplify the presentation, we only show the training loss iterating over images. A symmetrical loss $L_T$ that iterates over texts is also computed during the training process. The total training loss is $L = (L_I + L_T)/2$.

**Language Rewrites as Text Augmentation.** In Equation 1, the standard CLIP loss applies augmentation exclusively to images, leaving the text inputs unchanged throughout the whole training process. Recognizing this gap, we propose to generate text augmentations, denoted as $\text{aug}_T$, where $\text{aug}_T(x_T)$ is utilized as the input for $f_T$ instead of the original $x_T$. In the following subsections, we introduce the methodology employed to generate these text augmentations using LLMs, as well as the integration process during CLIP training. By addressing this gap, we aim to enhance the training process and expand the benefits of augmentation to the text inputs, leading to improved performance and a more comprehensive learning framework.

## 3.2 Meta-Input-Output Text Pair Generation

A recently uncovered property of autoregressive LLMs is In-Context Learning (ICL) [5, 39]. This property allows LLMs to learn a new task by conditioning on a few examples and then make predictions for a test input. To harness ICL for text rewriting, we first need to generate several

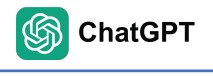

**Source Captions**
1. *white and red cheerful combination in the **bedroom** for a **girl***
2. *A **tourist** taking a **photograph** of **river** looking towards suspension **bridge** and **office***

   *...*

N. *tree **hollow** and **green leaves** of a **tree top** in summer*

**ChatGPT**

*"rewrite this image caption"*

**Target Captions**
1. *A bright and lively white-and-red color scheme in a **girl's bedroom**, creating a cheerful ambiance.*
2. ***Tourist** snaps **photo** of suspension **bridge** and **office** building across the river.*

   *...*

N. *Amidst lush **green leaves** on the top of a **tree**, a **hollow** creates a natural shelter, typical of **summer** foliage.*

Figure 1: Illustraion of using ChatGPT to generate meta-input-output pairs: we first sample source captions randomly from a few datasets. We then use prompts such as "Rewrite this image caption" to guide the ChatGPT model to generate rewritten target captions. The resulting target captions have different sentence structure than the source texts but, crucially, keep the major objects and subjects intact (in bold). These meta-input-output pairs are later used as contexts for ICL.

rewriting examples that can be used as examples in the prompt context. We name these examples meta-input-output text pairs. To this end, we explored different strategies for generating these pairs:

- **Rewriting with Chatbots.** We randomly sample texts from image-text datasets, and prompt ChatGPT[2] and Bard[1] web portals to generate target texts using a prompt such as "Rewrite this caption of an image vividly, and keep it less than thirty words:". Illustrations of this process can be found in Figure 1. Here we leverage the extremely powerful rewriting abilities of these models to provide modified captions that keep the essence of the original caption but change the style and details. This ensures that the semantics associated with the corresponding image do not change, which is important for representation learning purposes.
- **MSCOCO Sampling.** Multiple text descriptions for the same image are available in many existing image captioning datasets. To leverage this characteristic, we utilize the widely used MS-COCO dataset [36]. Within this dataset, each image is associated with five distinct text descriptions, which have been meticulously annotated by human workers. From this dataset, we randomly select a subset of images. For each selected image, we choose one description as the meta-input text and another as the meta-output text.
- **Human Rewriting.** We randomly sample several image-text pairs from various image-text datasets. To ensure diverse and varied textual variations, we engage human annotators and task them with rewriting the captions based on the content depicted in the corresponding observed images. This process results in the creation of meta-input-output pairs, consisting of the original text and the rewritten version by human annotators.

Through the utilization of diverse generation strategies, we acquire four distinct types (*ChatGPT*, *Bard*, *COCO*, and *Human*) of meta-input-output text pairs, which then serve as valuable examples within the input context for the In-Context Learning (ICL) framework. For each specific strategy, we randomly select 16 original captions from image-text datasets and generate target captions using that strategy, resulting in a total of 16 meta-input-output pairs. These pairs encompass a range of sources and variations, facilitating a comprehensive and diverse training experience for our framework.

### 3.3   Large scale Language Rewriting

Generating rewrites for hundreds of millions of texts using closed-source models like ChatGPT or Bard is impractical due to the significant financial and time costs associated with API usage. Therefore, to facilitate the rewriting of text samples within any given image-text dataset, we employ LLaMA [56]—an open-source state-of-the-art large language model known for its robust performance in text completion tasks. Despite not being fine-tuned with instructions, LLaMA exhibits exceptional ICL capabilities. Leveraging the meta-input-output text pairs generated as described in Section 3.2, we employ LLaMA's ICL ability to rewrite every text entry within the image-text dataset.

Given a text sample to be rewritten, we formulate a context input as the following three parts: Firstly, we include a sentence that informs the LLM about the task of rewriting image descriptions. This serves as an initial contextual clue for the LLM to understand the objective at hand. The second part of the context encompasses three examples sampled from the meta-input-output pairs described in Section 3.2.We randomly three distinct meta-input-output caption pairs from a specific strategy (e.g.,

**Context**

*Rewrite the following image descriptions:*

*white red cheerful combination in the bedroom for girl => A bright and lively white-and-red color scheme in a girl's bedroom, creating a cheerful ambiance.*

*A tourist taking a photograph of river looking towards suspension bridge and office => Tourist snaps photo of suspension bridge and office building across the river.*

*tree hollow and green leaves of a tree top in summer => Amidst lush green leaves on the top of a tree, a hollow creates a natural shelter, typical of summer foliage.*

*man driving a car through the mountains =>*

LLaMA
Text Completion

**Augmented Caption**

*A man confidently navigating a winding mountain road with breathtaking views.*

Figure 2: Illustration of our proposed in-context learning based strategy for language rewriting: The left box depicts the input context for LLaMA, responsible for text completion. The blue and green texts represent the meta-input-output text pairs, with blue indicating the input and green indicating the output. These pairs are defined in Section 3.2 and visualized in Fig. 1. The final blue line represents the text from the image-text datasets intended for rewriting. On the right box, we showcase the completion result generated by the open-source 7B parameter LLaMA model [56], which represents the rewritten version of the text of interest.

ChatGPT). Each pair is clearly separated by a "=>" symbol. These pairs provide explicit instances that showcase the desired rewriting behavior for the LLM to learn from. The additional random sampling process further enable the LLaMA model to generate more diverse text rewrites. Finally, the last part of the context includes the text sample that requires rewriting, followed by the separation symbol. This ensures that the LLM receives the specific text to be rewritten as part of its context input. By incorporating these three parts, we create a comprehensive context that guides the LLM in effectively generating diverse and contextually appropriate text rewrites.

Utilizing the constructed context input as a prompt, LLaMA exhibits its ability to perform text completion and generate rewritten versions of the corresponding text samples. This process is conducted for each text sample present in the pre-training image-text dataset. Specifically, we employ the LLaMA-7B model to generate four distinct rewrites for every text sample in the dataset, with each rewrite corresponding to one of the four different meta-input-output sources (*ChatGPT*, *Bard*, *COCO*, or *Human*). It takes 7 hours to generate one rewrite for the entire CC3M dataset on a 8 A100 GPU machine. By incorporating multiple sources and leveraging the capabilities of LLaMA, we ensure the generation of diverse and contextually relevant text rewrites for each text sample within the dataset.

### 3.4 LaCLIP: Training CLIP with Language Augmentations

Having generated $M$ different rewrites for each caption (in our case, $M = 4$), we are now able to bridge the augmentation gap between the image and text inputs, thereby presenting the training strategy for our LaCLIP framework. The key addition to the CLIP framework is the augmentation function for the text inputs, which can be easily implemented through a straightforward random sampling process, where we randomly select a text sample from either the original text or one of the generated rewrites:

$$\text{aug}_T(x_T) \sim \text{Uniform}([x_{T0}, x_{T1} \ldots, x_{TM}]) \tag{2}$$

Here $x_{Tk}$ is the $k^{th}$ rewritten sample for $x_T$. For the sake of simplicity, we denote the original text as $x_{T0}$. The training loss over the images becomes:

$$L_I = -\sum_{i=1}^{N} \log \frac{\exp\left(\text{sim}(f_I(\text{aug}_I(x_I^i)), f_T(\text{aug}_T(x_T^i)))/\tau\right)}{\sum_{k=1}^{N} \exp\left(\text{sim}(f_I(\text{aug}_I(x_I^i)), f_T(\text{aug}_T(x_T^k)))/\tau\right)}, \tag{3}$$

The only difference with the original CLIP training here is the additional text augmentation $\text{aug}_T$, and all other parts remains the same, which does not bring any additional computation or parameter overheads compared to original CLIP during training. By incorporating text augmentations into CLIP, we introduce variability and diversity into the training data, enabling the model to learn from both the original text and the augmented versions. This simple yet effective strategy enhances the training process and contributes to the overall performance and adaptability of the LaCLIP framework.

Table 1: Zero-shot transfer evaluation of different models. Performance on ImageNet and 15 common downstream datasets are reported. We highlight the best performance of each setting in **bold**. We see that regardless of the scale of the pre-training dataset, LaCLIP outperforms CLIP [44] and LaSLIP outperforms SLIP [41], by up to 8.2% absolute accuracy.

| Data | Model | Food-101 | CIFAR-10 | CIFAR-100 | SUN397 | Cars | Aircraft | DTD | Pets | Caltech-101 | Flowers | STL-10 | EuroSAT | RESISC45 | GTSRB | Country211 | Average | ImageNet |
|---|---|---|---|---|---|---|---|---|---|---|---|---|---|---|---|---|---|---|
| _Model Architecture: ViT-B/32_ | | | | | | | | | | | | | | | | | | |
| LAION-400M CLIP | CLIP | **79.9** | 91.8 | 72.0 | 64.6 | 77.0 | 15.8 | 49.9 | 84.8 | 89.3 | 64.4 | 95.3 | 43.2 | 60.6 | 36.9 | 14.5 | 62.7 | 62.0 |
| LAION-400M | LaCLIP | 79.7 | **92.4** | **73.0** | **64.9** | **81.9** | **20.8** | **55.4** | **87.2** | **91.8** | **70.3** | **97.3** | **50.6** | **61.5** | **49.4** | **16.0** | **66.1** | **64.4** |
| _Model Architecture: ViT-B/16_ | | | | | | | | | | | | | | | | | | |
| CC3M | CLIP | 10.3 | 54.9 | 21.8 | 25.0 | 0.8 | 1.4 | 10.5 | 12.8 | 43.3 | 10.2 | 77.6 | 14.1 | 19.1 | **6.9** | 0.6 | 20.6 | 15.8 |
| CC3M | LaCLIP | **14.2** | **57.1** | **27.5** | **35.1** | **1.6** | **1.6** | **16.6** | **15.6** | **52.7** | **14.7** | **86.2** | **15.0** | **24.3** | 6.4 | **1.0** | **24.6** | **21.5** |
| CC12M | CLIP | 50.8 | 64.9 | 38.5 | 44.7 | 24.1 | 2.4 | 19.4 | 64.1 | 77.4 | 33.2 | 91.0 | 20.1 | 38.9 | 7.3 | 5.1 | 38.8 | 40.2 |
| CC12M | LaCLIP | **60.7** | **75.1** | **43.9** | **57.0** | **36.3** | **5.6** | **31.0** | **72.4** | **83.3** | **39.9** | **95.1** | **27.3** | **44.3** | **12.7** | **8.9** | **46.2** | **48.4** |
| CC12M | SLIP | 52.5 | 80.7 | 46.3 | 48.8 | 24.9 | 2.3 | 25.1 | 58.6 | 77.6 | 29.2 | 89.1 | **25.8** | 36.6 | 6.0 | 5.7 | 40.6 | 42.1 |
| CC12M | LaSLIP | **62.9** | **82.0** | **50.2** | **59.6** | **32.2** | **4.4** | **30.1** | **70.6** | **82.4** | **37.4** | **95.0** | 20.4 | **45.6** | **10.1** | **9.2** | **46.1** | **49.7** |
| RedCaps | CLIP | 81.5 | 70.4 | 39.9 | 33.2 | 19.2 | 1.9 | 19.7 | **82.7** | 72.8 | 53.9 | **92.8** | 23.3 | 33.6 | **8.3** | 6.2 | 42.6 | 42.9 |
| RedCaps | LaCLIP | **85.0** | **74.8** | **40.7** | **40.3** | **21.3** | **2.2** | **23.9** | 78.2 | **76.4** | **59.0** | 91.4 | **27.1** | **41.3** | 5.6 | **7.6** | **45.0** | **46.2** |
| LAION-400M | CLIP | 85.5 | 93.0 | 71.7 | 66.8 | 83.5 | 16.7 | 52.8 | 90.1 | 91.2 | 63.9 | 97.3 | 42.4 | 63.3 | **46.2** | 17.8 | 65.5 | 67.0 |
| LAION-400M | LaCLIP | **86.5** | **93.5** | **73.9** | **67.9** | **87.1** | **24.2** | **58.9** | **90.9** | **92.4** | **73.1** | **98.4** | **48.3** | **65.8** | 46.1 | **19.6** | **68.4** | **69.3** |

# 4 Experiments

**Datasets.** Our experiments were conducted on four different image-text datasets at different scale: Conceptual Captions 3M (CC3M) [51], Conceptual Captions 12M (CC12M) [7], RedCaps [15], and LAION-400M[49]. RedCaps is a 12M-instance dataset collected exclusively from Reddit, potentially exhibiting distinct distributions compared to other datasets. The majority of our ablation studies were performed on the CC12M dataset. We evaluate all the models on ImageNet and 15 common downstream datasets like Food101 [4], SUN397 [59] and FGVCAircraft [38]. Appendix A contains the details for all datasets.

**Training Parameters.** For most of our experiments on CC3M, CC12M, and RedCaps, we utilized the ViT-B/16 architecture [17] and trained the models with a batch size of 8,192 and the AdamW optimizer [26]. Additionally, we explored the ViT-L/16 and ViT-S/16 architectures in ablation studies. For LAION-400M, we used both the ViT-B/32 and ViT-B/16 architecture with a batch size of 32,768, and followed the exact training setup outlined in [44], training the model for 32 epochs. Appendix B contains a detailed breakdown of our training hyperparameters.

**Evaluation Setup.** We consider three evaluation metrics for the trained models: Zero-Shot (**ZS**) classification accuracy, Few-Shot (**FS**) classification accuracy and Linear Probing (**LP**) accuracy. For zero-shot classification, we adopt the same prompt templates as described in the CLIP paper [44]. The class text embeddings are used to compute distance with the image feature, and images are classified to class with the shortest distance. For few-shot classification, we follow the set up in [44] and perform 5-way 5-shot classification with a weighted kNN classifier on top of the frozen features. For linear probing, following [44, 18], we freeze the pre-trained image encoder and extract features for every image in the downstream dataset. We then train a linear classifier using L-BFGS optimizer on top of the extracted features. ZS and LP are evaluated on both ImageNet (**IN**) and 15 Downstream (**DS**) datasets. FS are evaluated on the same downstream datasets. In the ablation studies we report the perforamnce on **IN** and the mean on **DS**.

## 4.1 Zero-shot Evaluation

We provide a comprehensive analysis of the zero-shot transfer performance on ImageNet and downstream datasets in Table 1. Remarkably, across all pretrained datasets, our LaCLIP approach achieves a significant performance improvement over the baseline CLIP model on both ImageNet and down-

Table 2: Few-shot transfer evaluation of different models. We report 5-way, 5-shot classification accuracy for all downstream datasets. We highlight the best performance of each setting in **bold**. Similar to zero-shot, in nearly all cases, pre-training using language rewrites outperforms vanilla pre-training.

| Data | Model | Food-101 | CIFAR-10 | CIFAR-100 | SUN397 | Cars | Aircraft | DTD | Pets | Caltech-101 | Flowers | STL-10 | EuroSAT | RESISC45 | GTSRB | Country211 | Average |
|---|---|---|---|---|---|---|---|---|---|---|---|---|---|---|---|---|---|
| | | | | | | | *Model Architecture: ViT-B/32* | | | | | | | | | | |
| LAION-400M | CLIP | 92.5 | 87.2 | 89.0 | 98.0 | 98.5 | 78.9 | 87.4 | 94.5 | 99.2 | 99.0 | 96.1 | **82.8** | **94.3** | 79.8 | 49.7 | 88.5 |
| | LaCLIP | **93.5** | **91.0** | **90.7** | **98.2** | **99.1** | **82.2** | **87.5** | **95.7** | **99.4** | **99.2** | **97.2** | 80.1 | 94.2 | **80.4** | **52.2** | **89.4** |
| | | | | | | | *Model Architecture: ViT-B/16* | | | | | | | | | | |
| CC3M | CLIP | 67.6 | 64.2 | 73.6 | 94.1 | 54.4 | 46.1 | 74.4 | 76.7 | 93.3 | 94.3 | 84.6 | **81.4** | **87.1** | 66.9 | 37.3 | 73.1 |
| | LaCLIP | **70.0** | **69.1** | **76.8** | **95.2** | **57.6** | **49.2** | **75.8** | **77.4** | **95.2** | **95.0** | **89.5** | 81.1 | 85.5 | **71.0** | 37.3 | **75.0** |
| CC12M | CLIP | 87.0 | 77.5 | 82.1 | 97.2 | 90.9 | 62.0 | 83.3 | 91.1 | 98.2 | 97.6 | 92.6 | **83.4** | 91.2 | 70.6 | 44.3 | 83.3 |
| | LaCLIP | **89.9** | **81.3** | **85.0** | **98.0** | **95.3** | **68.1** | **84.9** | **93.4** | **98.9** | **98.4** | **95.9** | 83.0 | **92.4** | **76.4** | **46.7** | **85.8** |
| | SLIP | 87.6 | 79.2 | 83.0 | 97.5 | 85.6 | 56.4 | 85.8 | 88.1 | 97.7 | 97.1 | 92.5 | **84.9** | 91.0 | 62.4 | 43.0 | 82.1 |
| | LaSLIP | **90.5** | **84.9** | **86.6** | **98.1** | **91.6** | **61.0** | **86.7** | **89.8** | **98.7** | **97.8** | **94.2** | 84.0 | **92.8** | **65.8** | **45.4** | **84.5** |
| RedCaps | CLIP | 94.4 | 80.6 | 85.3 | 95.9 | 88.5 | 54.5 | **82.6** | **94.5** | 97.8 | 99.0 | 94.8 | 84.9 | 91.3 | 75.3 | 40.6 | 84.0 |
| | LaCLIP | **95.8** | **81.4** | **85.4** | **96.2** | **90.9** | **58.8** | 82.4 | 94.1 | **98.0** | **99.2** | **95.6** | **86.2** | **92.1** | **76.5** | **42.6** | **85.0** |
| LAION-400M | CLIP | 95.0 | 90.1 | 90.7 | 98.2 | 99.2 | 80.8 | 88.7 | 96.2 | 99.5 | 99.4 | 97.1 | **84.5** | 95.0 | 77.7 | 55.1 | 89.8 |
| | LaCLIP | **95.8** | **92.7** | **91.9** | **98.4** | **99.5** | **86.1** | **89.0** | **97.1** | **99.6** | **99.5** | **98.1** | 82.9 | 95.0 | **80.9** | **57.9** | **91.0** |

stream datasets. For instance, when training models on the CC12M dataset, our LaCLIP method achieves over 8% improvement in absolute top-1 accuracy on ImageNet and 7% improvement on average over the other downstream datasets. LaCLIP and CLIP share the exact same amount of parameters and computation cost during training.

**Adaptability to other methods.** It is noteworthy that LaCLIP is compatible with other techniques intended to enhance CLIP's performance. Once the augmented texts are generated, integrating LaCLIP into any CLIP-based framework can be achieved seamlessly without incurring additional computational or memory overhead. As demonstrated in Table 1, we applied language augmentation to the SLIP framework and yield LaSLIP, resulting in significant performance improvements across all evaluation metrics. Notably, even though SLIP already incorporates additional self-supervision to enhance CLIP's performance, our proposed language augmentation further boosts its effectiveness. This showcases the generalization capability of our proposed text augmentation strategy.

**Generalization to Larger Datasets.** Consistently, the results highlight the substantial margin by which LaCLIP outperforms CLIP across various datasets. Noteworthy is the scalability of our method with dataset size, as it demonstrates improvement even when trained on the massive LAION-400M dataset, which contains hundreds of millions of data points[1]. These findings suggest that our LaCLIP approach can be seamlessly integrated as a plug-and-play component for training vision-language foundation models.

## 4.2 Few-Shot & Linear-Probing

We present the 5-way 5-shot classification performance in Table 2 and the linear-probing performance in Table 3. Our approach consistently outperforms vanilla CLIP or SLIP in the vast majority of cases. Interestingly, SLIP performs worse than vanilla CLIP in the few-shot setting, despite introducing additional self-supervision from the image side. However, by incorporating our proposed language augmentation strategy, SLIP's few-shot performance improves, surpassing vanilla CLIP. This result highlights the effectiveness of text augmentations in the few-shot setting.

Furthermore, it is important to highlight that the improvements observed in the few-shot and linear-probing results are solely achieved through the utilization of image encoders. This demonstrates the efficacy of our proposed text augmentation approach in enhancing not only the joint image-text embedding space, which aligns image and text features more effectively, but also the quality of the

---

[1]The version used in our experiment contains ∼340M samples, slightly less than original due to link rot. We use OpenCLIP implementation ( https://github.com/mlfoundations/open_clip) and achieves 62.0% ImageNet zero-shot accuracy for CLIP, comparable to their model with 62.9% trained on the full dataset.

Table 3: Linear Probing comparison of different models. Performances on ImageNet and 15 common downstream datasets are reported. We highlight the best performance of each setting in **bold**. Similar to ZS and FS, linear probe performance for our approach is almost always better than that of vanilla CLIP or SLIP.

| Data | Model | Food-101 | CIFAR-10 | CIFAR-100 | SUN397 | Cars | Aircraft | DTD | Pets | Caltech-101 | Flowers | STL-10 | EuroSAT | RESISC45 | GTSRB | Country211 | Average | ImageNet |
|---|---|---|---|---|---|---|---|---|---|---|---|---|---|---|---|---|---|---|
| | | *Model Architecture: ViT-B/32* | | | | | | | | | | | | | | | | |
| LAION-400M | CLIP | **85.8** | 95.8 | 83.6 | 75.1 | 89.2 | 54.3 | **79.7** | 86.9 | 94.5 | 96.8 | 97.9 | **96.3** | 93.5 | 88.6 | **23.1** | 82.7 | 74.6 |
| | LaCLIP | 85.1 | **96.2** | **84.2** | **75.6** | **90.1** | **56.1** | 79.6 | **89.1** | **94.8** | **97.7** | **98.4** | 95.8 | **93.6** | 88.6 | 22.9 | **83.2** | 75.3 |
| | | *Model Architecture: ViT-B/16* | | | | | | | | | | | | | | | | |
| CC3M | CLIP | 62.6 | 86.8 | 68.1 | 58.5 | **32.8** | 40.9 | 63.4 | 69.6 | 82.0 | 89.4 | 91.7 | **95.9** | 89.0 | 71.9 | **13.3** | 67.7 | 54.5 |
| | LaCLIP | **63.8** | **87.7** | **69.5** | **60.2** | 32.4 | **42.7** | **64.0** | **71.1** | **83.3** | **90.2** | **93.4** | 95.8 | **89.7** | **74.6** | 13.2 | **68.8** | 56.5 |
| CC12M | CLIP | 81.6 | 93.8 | 79.3 | 72.0 | 75.1 | 52.6 | 75.6 | 86.2 | 92.2 | 95.3 | 97.3 | **96.7** | **93.1** | 80.6 | 19.7 | 79.4 | 70.3 |
| | LaCLIP | **82.9** | **94.7** | **79.7** | **73.8** | **79.9** | 54.5 | **75.7** | **87.7** | **93.0** | **96.4** | **98.0** | 96.4 | 93.0 | **81.9** | 19.7 | **80.5** | 72.3 |
| | SLIP | 84.4 | 94.2 | 79.1 | 73.5 | 74.2 | **54.6** | 76.5 | **86.1** | 92.7 | 95.7 | 97.6 | 96.8 | **93.7** | 74.0 | 20.6 | 79.6 | 73.2 |
| | LaSLIP | **85.2** | **94.6** | **80.8** | **75.1** | **77.0** | 53.8 | **78.5** | 85.6 | **93.7** | **96.5** | **97.9** | 96.8 | 93.5 | **76.1** | **21.1** | **80.4** | 74.4 |
| RedCaps | CLIP | 89.1 | 94.1 | **78.8** | 65.6 | 74.0 | 52.5 | 73.2 | **91.5** | 91.4 | 97.7 | **98.0** | 96.3 | **93.5** | 80.8 | 17.0 | 79.6 | 71.8 |
| | LaCLIP | **90.1** | **94.3** | 78.5 | **66.6** | **77.6** | **53.6** | **73.9** | 90.8 | **91.5** | **97.9** | 97.6 | **96.6** | 92.7 | 80.8 | **17.2** | **80.0** | 71.9 |
| LAION-400M | CLIP | 90.5 | **96.9** | 85.0 | 78.1 | 92.1 | 57.2 | 80.0 | 90.9 | 95.7 | 98.0 | 98.7 | **96.7** | **94.7** | **90.3** | 27.0 | 84.8 | 78.6 |
| | LaCLIP | **90.7** | 96.7 | **85.5** | **78.7** | **92.8** | **63.1** | **81.3** | **92.8** | **96.2** | **98.8** | **99.1** | 96.4 | 94.6 | 89.5 | **27.5** | **85.6** | 79.9 |

image representations themselves. This improvement is realized by pairing images with a wider range of diverse texts, providing richer and more varied supervision to the vision encoder. As a result, the vision encoder is able to learn more generalizable and robust image representations that can be effectively utilized across a range of downstream tasks. To further emphasize the effectiveness of our approach, we provide detailed t-SNE visualizations of the learned features with different approaches in Appendix F. These visualizations qualitatively demonstrate that features learned with LaCLIP exhibit clearer class boundaries across different classes, reinforcing the enhanced discriminative power of our approach.

## 4.3 Ablation Studies

**Varying Augmentation Strategies.** Table 4 presents a comparison of our proposed LLM-based text augmentation strategy with existing language augmentation methods, namely EDA [58], which involves word-level operations such as synonym replacements, and back translation [50], which translates the text to another language and then back to the original language. Although these simpler augmentation techniques demonstrate some improvements, the enhancements achieved are relatively marginal, especially when compared to our proposed LLM-based approach. This disparity can be attributed to the significantly higher diversity in sentence structures that our method achieves. In Appendix D, we provide further details and qualitative comparisons of the various text augmentation techniques using different methods.

**Scaling with Number of Augmentations.** Figure 3 demonstrates the impact of the number of augmented texts per sample in our LaCLIP approach on ImageNet zero-shot accuracy. A value of 0 augments corresponds to vanilla CLIP without any text augmentation. The results clearly demonstrate that simpler augmentation strategies exhibit poor scalability as the number of augments increases. This is attributed to their limited diversity. Conversely, our LLM-based text augmentation consistently improves performance as more augmentations are added.

**Different Meta-Input-Output Pair for ICL.** In order to assess the impact of different meta-input-output caption pairs used for LLaMA ICL, we conducted an experiment by disentangling LaCLIP and training four models. Each model was trained using only the ground truth caption combined with a specific type of LLM augmented version, using a particular meta-input-output pair (ChatGPT, Bard, COCO, or human). The performance comparison between these models is presented in Table 5. It can be observed that different strategy yield similar performance, with the model trained with augmentations using the Human pair slightly outperforming the others. We conjecture the reason

Table 4: Performance comparison of CLIP training with different text augmentations.

| Augment | ZS | | FS | LP | |
|---|---|---|---|---|---|
| | DS | IN | | DS | IN |
| N/A (CLIP) | 38.8 | 40.2 | 83.3 | 79.4 | 70.3 |
| EDA [58] | 40.6 | 41.2 | 83.4 | 79.4 | 70.5 |
| Back Trans [50] | 40.4 | 41.6 | 83.9 | 79.8 | 70.7 |
| LLM (Ours) | **46.2** | **48.4** | **85.8** | **80.5** | **72.3** |

Table 5: Ablation of CLIP trained with augmented texts prompted by different meta-input-output pairs.

| Source | ZS | | FS | LP | |
|---|---|---|---|---|---|
| | DS | IN | | DS | IN |
| ChatGPT | 42.3 | 44.5 | 84.8 | 79.8 | 71.2 |
| Bard | 41.7 | 44.8 | 85.0 | 79.6 | 71.2 |
| MSCOCO | 42.1 | 44.6 | 84.8 | 79.8 | 71.3 |
| Human | 43.0 | 45.1 | 84.8 | 79.9 | 71.3 |

Table 6: Ablation on different network architectures.

| Backbone | Method | ZS | | FS | LP | |
|---|---|---|---|---|---|---|
| | | DS | IN | | DS | IN |
| ViT-S/16 | CLIP | 36.3 | 36.9 | 82.2 | 77.0 | 67.1 |
| | LaCLIP | 44.1 | 46.3 | 84.5 | 78.0 | 69.1 |
| ViT-B/16 | CLIP | 38.8 | 40.2 | 83.3 | 79.4 | 70.3 |
| | LaCLIP | 46.2 | 48.4 | 85.8 | 80.5 | 72.3 |
| ViT-L/16 | CLIP | 42.6 | 44.0 | 85.1 | 81.3 | 72.9 |
| | LaCLIP | 46.6 | 49.1 | 86.8 | 81.9 | 73.7 |

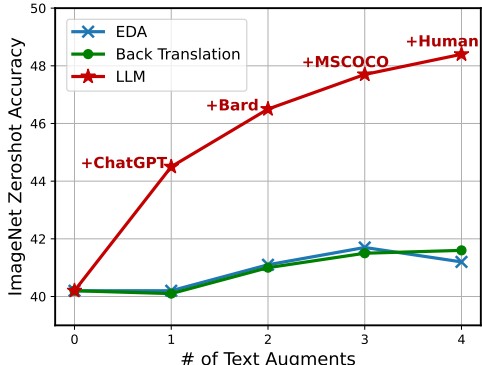

Figure 3: ImageNet zero-shot accuracy with different num of text augments.

Table 7: Performance comparison of CLIP, LaCLIP and LaCLIP-MT on different datasets.

| Dataset | Method | ZS | | FS | LP | |
|---|---|---|---|---|---|---|
| | | DS | IN | | DS | IN |
| CC12M | CLIP | 38.8 | 40.2 | 83.3 | 79.4 | 70.3 |
| | LaCLIP | **46.2** | 48.4 | 85.8 | 80.5 | 72.3 |
| | LaCLIP-MT | 45.2 | **49.0** | 85.8 | **80.6** | **72.4** |
| RedCaps | CLIP | 42.6 | 42.9 | 84.0 | 79.6 | 71.8 |
| | LaCLIP | 45.0 | 46.2 | 85.0 | 80.0 | 71.9 |
| | LaCLIP-MT | **46.1** | **48.1** | **85.3** | **80.3** | **72.4** |

is that during the meta-input-output generation process, humans have the advantage of viewing the corresponding image, which allows them to generate more accurate and diverse rewrites.

**Different Backbone Architecture.** We further investigate the performance of LaCLIP using different backbone architectures. Table 6 summarize the results obtained with ViT-S/16, ViT-B/16, and ViT-L/16. We observe that LaCLIP scales with model size and consistently outperforms vanilla CLIP across all network architectures. These findings highlight the effectiveness of LaCLIP in improving the performance of CLIP models, irrespective of the underlying backbone architecture.

**Comparison with Pre-trained Text Encoder.** To deepen the comprehension on whether LaCLIP outperforms vanilla CLIP simply due to a better text encoder, we conducted experiments comparing LaCLIP with CLIP models trained with a pre-trained text encoder. Here we employed the BERT model as the pre-trained text encocder. The experiment result in Table 8 demonstrates that fine-tuning based on the pre-trained text encoder exhibits some improvements, whereas freezing the pre-trained text encoder weights substantially degrades performance.

Table 8: Performance comparison of LaCLIP and CLIP models trained with different pre-trained text encoder configurations.

| Method | Text Encoder | | ZS | | FS | LP | |
|---|---|---|---|---|---|---|---|
| | Pre-train | Freeze | DS | IN | | DS | IN |
| CLIP | ✗ | ✗ | 38.8 | 40.2 | 83.3 | 79.4 | 70.3 |
| | ✔ | ✗ | 42.1 | 42.9 | 83.6 | 79.5 | 70.4 |
| | ✔ | ✔ | 24.5 | 23.2 | 80.3 | 74.9 | 66.0 |
| LaCLIP | ✗ | ✗ | **46.2** | **48.4** | **85.8** | **80.5** | **72.3** |

This observation aligns with the findings in LiT [63]. In contrast, LaCLIP consistently outperforms all configurations with a pre-trained text encoder, underscoring the benefit and necessity for explicit sentence augmentation strategies.

## 5 Multi-Text Training Loss with LaCLIP

It is important to highlight that once we have generated multiple text augmentations, and with a slight tolerance for computational cost, we can create multi-positive training pairs for each training iteration. These pairs are formed by pairing each image with not only the original text but also with all the rewritten versions of the text. By adopting this approach, we introduce a Multi-Text version

of LaCLIP, referred to as LaCLIP-MT, which incorporates a multi-positive contrastive training loss. The training process iterates through all the images in the following manner:

$$L_{I*} = -\frac{1}{M} \sum_{i=1}^{N} \sum_{j=0}^{M} \log \frac{\exp\left(\text{sim}(f_I(\text{aug}_I(x_I^i)), f_T(x_{Tj}^i))/\tau\right)}{\sum_{k=1}^{N} \exp\left(\text{sim}(f_I(\text{aug}_I(x_I^i)), f_T(x_{Tj}^k))/\tau\right)}, \quad (4)$$

Since each text can still be paired with a single image, the training loss that iterates through all the texts remains unchanged, with the only difference being that it now iterates over all of the texts instead of just the augmented ones. Consequently, the final training loss is given by the average of the image loss ($L_{I*}$) and the text loss ($L_T$), resulting in $L = (L_{I*} + L_T)/2$.

In order to showcase the efficacy of the multi-positive contrastive training loss in boosting the performance of LaCLIP, we conducted additional experiments on the CC12M and RedCaps datasets. The results of these experiments are summarized in Table 7, which compares the performance of LaCLIP-MT with both LaCLIP and vanilla CLIP. The results clearly indicate that LaCLIP-MT could further improve upon LaCLIP across most metrics. By allowing each image to be paired with all of the diverse texts describing its content, LaCLIP-MT leverages the additional and richer supervision from the language modality to enhance the formation of image-text embeddings. This improvement highlight the benefits of the multi-positive contrastive training loss in facilitating better alignment between images and diverse text descriptions.

# 6 Conclusion, Limitations and Broader Impact

**Conclusion.** We have introduced LaCLIP, a straightforward yet highly effective CLIP training strategy that incorporates text augmentations through text rewriting, leveraging the in-context learning capabilities of LLMs. Through this simple and versatile approach, we have demonstrated significant improvements in the performance of CLIP embeddings across various pre-training scales and datasets. Additionally, we have proposed a novel multi-text training loss to further enhance the training process. As LLMs continue to improve in performance and in-context learning capabilities, our approach stands to directly benefit from these advancements.

**Limitations.** While the training process itself does not entail any additional memory or computation overhead compared to vanilla CLIP, the process of generating text rewrites using LLMs can be computationally expensive, requiring significant GPU resources and taking hours for large datasets. Additionally, the quality of the rewritten text generated by LLaMA is not filtered, which may result in some irrelevant details that do not align well with the corresponding images. This misalignment could impact the transferability of the learned embeddings to downstream tasks. To address these limitations, future work could focus on developing more efficient methods for generating text rewrites using LLMs, reducing the computational burden without sacrificing performance. Furthermore, techniques for filtering the rewritten texts could be explored, aiming to retain only the most relevant and accurate versions while discarding those with misleading details. This would enable the model to learn a better embedding space that is robust and transferable across different downstream datasets, improving overall performance and alignment between vision and text encoders.

**Broader Impact.** We propose a general text augmentation strategy that can generate diverse rewrites for any given text. This strategy not only improves the performance of vision-language models but also has the potential to enhance models in pure natural language processing tasks, such as language understanding and reasoning. On the other hand, we acknowledge that LLMs are trained on large-scale web data, which may contain factual errors and hallucinations. Consequently, the rewritten versions of texts may also inherit these limitations. Therefore, we encourage researchers to implement additional data filtering methods before deploying these models in real-world scenarios. Additionally, the current LLM-based rewriting strategy requires significant GPU/TPU computation, which can contribute to a higher carbon footprint. However, it is also possible that such rewriting strategy can significantly reduce the number of training iterations for larger models to reach similar performances as vanilla CLIP.

## Acknowledgements

We would like to thank Mathilde Caron for the insightful early manuscript review, and the anonymous reviewers for their helpful comments and suggestions. Additionally, we appreciate the support and discussions with the VisCAM team at Google Research.

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

# Appendices

We provide additional experiment details, results and analysis in the supplementary material.

## A. Dataset Details

### A.1 Pre-training Datasets

We utilized four image-text pre-training datasets of varying scales, namely CC3M [51], CC12M [7], RedCaps [15], and LAION-400M [49], to train both CLIP and LaCLIP models. Additionally, we trained SLIP and LaSLIP models on the CC12M dataset. It is important to note that due to image link rot within the datasets, the versions we obtained may have slightly fewer images compared to the original versions. As a result, there may be slight performance differences when compared to models trained on the full image versions. Below are detailed descriptions of the four pre-training datasets:

**CC3M [51]:** This dataset comprises 3.3 million image-text pairs extracted from 5 billion webpages. The image descriptions are derived from the HTML alt-text attribute. The version we used consists of 2.8 million unique samples.

**CC12M [7]:** With a similar procedure as CC3M, CC12M consists of 12.4 million image-text pairs. The filters used in this dataset are more relaxed, resulting in a wider range of topics and visual concepts, making it more reflective of real-world scenarios. The version we acquired contains 10.0 million samples.

**RedCaps [14]:** RedCaps encompasses 12.0 million image-caption pairs gathered exclusively from Reddit across 350 subreddits. The captions are sourced from Reddit instead of HTML alt-text. The version we acquired includes 11.7 million unique samples.

**LAION-400M [49]:** This dataset is constructed by processing and filtering the Common Crawl dataset. The original version contains 413 million unique samples, while the version we obtained consists of 340 million samples.

For all datasets, we resized the images such that the shorter side measured 256 pixels.

### A.2 Downstream Datasets

We conducted evaluations on our pre-trained model using both ImageNet [13] and 15 widely-used downstream datasets. To prepare the downstream datasets, we utilized torchvision and VISSL [21]. The detailed information about the downstream datasets can be found in Table A1.

## B. Implementation Details

**Encoders** We employed the standard ViT-S/16, ViT-B/16, ViT-B/32 and ViT-L/16 architectures from [17, 55] as our vision encoders. Specifically, ViT-B/16 is used on all CC3M, CC12M and RedCaps datasets. ViT-B/32 is used on LAION-400M. ViT-S/16 and ViT-L/16 are used on CC12M. Following the approach in SLIP [41], we utilized the smallest text encoder from CLIP [44]. Our tokenizer was consistent with CLIP, having a vocabulary size of $49,408$ and a maximum context length of 77. Further details about the encoders can be found in Table A2.

**Hyper-Parameters** Table A3 provides an overview of the pre-training hyperparameters used for CLIP on all datasets. Following [44, 41], we perform RandomResizedCrop augmentation for the images. For SLIP training, the learning rate was set to $3 \times 10^{-3}$, weight decay was set to $0.1$, and all other parameters remained the same. Further details can be found in Table A4. The pre-training process was conducted on four machines with eight A100 GPUs each.

**Zero-shot Classification** We follow a similar prompt ensemble strategy as described in [44] and employ the same set of prompting templates. For each class name, we compute the average text embedding across all templates. These averaged embeddings are then used to calculate the similarity between each test image and the class embeddings. Specifically, for zero-shot evaluation on ImageNet, models trained on the LAION-400M dataset use the exact 80 prompts provided by [44] to ensure a

Table A1: Details of the downstream classification datasets.

| Dataset | Metric | Categories | Train Size | Test Size |
|---------|--------|------------|------------|-----------|
| Food-101 [4] | Accuracy | 101 | 75,750 | 25,250 |
| CIFAR-10 [29] | Accuracy | 10 | 50,000 | 10,000 |
| CIFAR-100 [29] | Accuracy | 100 | 50,000 | 10,000 |
| SUN397 [59] | Accuracy | 397 | 19,850 | 19,850 |
| Stanford Cars [28] | Accuracy | 196 | 8,144 | 8,041 |
| FGVC Aircraft [38] | Mean per class | 100 | 6,667 | 3,333 |
| DTD [11] | Accuracy | 47 | 3,760 | 1,880 |
| Oxford Pets [43] | Mean per class | 37 | 3,680 | 3,669 |
| Caltech-101 [19] | Mean per class | 102 | 3,060 | 6,085 |
| Oxford Flowers [42] | Mean per class | 102 | 2,040 | 6,149 |
| STL-10 [12] | Accuracy | 10 | 1,000 | 8,000 |
| EuroSAT [23] | Accuracy | 10 | 10,000 | 5,000 |
| RESISC45 [9] | Accuracy | 45 | 25,200 | 6,300 |
| GTSRB [53] | Accuracy | 43 | 26,640 | 12,630 |
| Country211 [44, 54] | Accuracy | 211 | 42,200 | 21,100 |

Table A2: Encoder details.

| Model | Patch size | Input resolution | Embedding dimension | Vision Transformer | | | Text Transformer | | | Vocab size | Text length |
|-------|-----------|------------------|---------------------|-------|-------|-------|--------|-------|-------|-----------|-------------|
| | | | | Layers | Width | Heads | Layers | Width | Heads | | |
| ViT-S/16 | 16 | 224 | 512 | 12 | 384 | 12 | 12 | 512 | 8 | | |
| ViT-B/16 | 16 | 224 | 512 | 12 | 768 | 12 | 12 | 512 | 8 | 49,408 | 77 |
| ViT-B/32 | 32 | 224 | 512 | 12 | 768 | 12 | 12 | 512 | 8 | | |
| ViT-L/16 | 16 | 224 | 512 | 24 | 1024 | 16 | 12 | 512 | 8 | | |

fair comparison. For models trained on other datasets, we use a subset of 7 templates recommended by [44] to expedite the evaluation process.

**Few-shot Classification** Following the settings in [18], we evaluate the 5-way 5-shot performance across 15 downstream datasets. We use Prototypical Networks [52] as classifier on top of the features extracted from vision encoders without data augmentation. Only Resize followed by CenterCrop is applied here for all images. We evaluate each model for 600 randomly sampled episodes, and for each episode, images are sampled from the combination of training, validation and testing sets. We always sample 15 images for each class as query set. The mean accuracy across all episodes are reported in the main paper, and we also report the 95% confidence interval in the appendix.

**Linear-Probing** For linear probing on ImageNet, we keep the image encoder frozen and train a Linear Classifier on the extracted features. The only augmentation applied is RandomHorizontalFlip. We sweep the base learning rate across the range of [0.002, 0.005, 0.01, 0.015, 0.02, 0.03, 0.05] and report the best performance achieved. The learning rate is scaled linearly based on the actual batch size, following the approach outlined in [20]. Details of all other hyperparameters can be found in Table A5. For linear probing on all other downstream datasets, we train a logistic regression layer on top of the frozen features extracted from the vision encoders, without applying any data augmentation. The model is optimized using L-BFGS with Scikit-learn, and the maximum number of iterations is set to 500. To determine the optimal $\ell_2$ regularization term for each model and dataset, we perform a sweep across 45 steps that are logarithmically spaced ranging from $10^{-6}$ to $10^5$ on the validation set. For the final results, we fit the model on the combined training and validation sets and report the performance on the separate test set.

## C. Meta-input-output Details

### C.1 Meta-input-output Pairs

Here we provide the exact 16 meta-input-output pairs we used as templates for all four set ups: ChatGPT, Bard, Human and MSCOCO, described in Section 3.2. We use 'Source' to represent the meta-input text we sampled from the image-text datasets, and use 'Target' to represent the meta-output text generated by each of the strategies. Note the meta-input-output pairs showed in Figure 1 and Figure 2 in the main text are for illustration only, please refer to this section for the real pairs used in the experiments.

Table A3: Detailed pre-training hyper-parameters for CLIP training on all four image-text datasets.

(a) Pre-training hyper-parameter on CC3M.

| Config | Value |
|---|---|
| Batch size | $8,192$ |
| Optimizer | AdamW [37] |
| Learning rate | $1 \times 10^{-3}$ |
| Weight decay | 0.5 |
| Adam $\beta$ | $\beta_1, \beta_2 = (0.9, 0.98)$ |
| Adam $\epsilon$ | $1 \times 10^{-8}$ |
| Total epochs | 40 |
| Warm up epochs | 1 |
| Learning rate schedule | cosine decay |

(b) Pre-training hyper-parameter on CC12M.

| Config | Value |
|---|---|
| Batch size | $8,192$ |
| Optimizer | AdamW [37] |
| Learning rate | $1 \times 10^{-3}$ |
| Weight decay | 0.5 |
| Adam $\beta$ | $\beta_1, \beta_2 = (0.9, 0.98)$ |
| Adam $\epsilon$ | $1 \times 10^{-8}$ |
| Total epochs | 35 |
| Warm up epochs | 1 |
| Learning rate schedule | cosine decay |

(c) Pre-training hyper-parameter on RedCaps.

| Config | Value |
|---|---|
| Batch size | $8,192$ |
| Optimizer | AdamW [37] |
| Learning rate | $1 \times 10^{-3}$ |
| Weight decay | 0.5 |
| Adam $\beta$ | $\beta_1, \beta_2 = (0.9, 0.98)$ |
| Adam $\epsilon$ | $1 \times 10^{-8}$ |
| Total epochs | 30 |
| Warm up epochs | 1 |
| Learning rate schedule | cosine decay |

(d) Pre-training hyper-parameter on LAION-400M.

| Config | Value |
|---|---|
| Batch size | $32,768$ |
| Optimizer | AdamW [37] |
| Learning rate | $5 \times 10^{-4}$ |
| Weight decay | 0.2 |
| Adam $\beta$ | $\beta_1, \beta_2 = (0.9, 0.98)$ |
| Adam $\epsilon$ | $1 \times 10^{-6}$ |
| Total epochs | 32 |
| Warm up iterations | $2,000$ |
| Learning rate schedule | cosine decay |

Table A4: SLIP hyper-parameters.

| Config | Value |
|---|---|
| Batch size | $8,192$ |
| Optimizer | AdamW [37] |
| Learning rate | $3 \times 10^{-3}$ |
| Weight decay | 0.1 |
| Adam $\beta$ | $\beta_1, \beta_2 = (0.9, 0.98)$ |
| Adam $\epsilon$ | $1 \times 10^{-8}$ |
| Total epochs | 35 |
| Warm up epochs | 1 |
| Learning rate schedule | cosine decay |

Table A5: Detailed hyper-parameters on Linear-Probing on ImageNet.

| Config | Value |
|---|---|
| Batch size | $1,024$ |
| Optimizer | SGD |
| Base learning rate | $sweep$ |
| Weight decay | 0 |
| Momentum | 0.9 |
| Training epochs | 90 |
| Learning rate schedule | cosine decay |

**ChatGPT:**

1. **Source:** white and red cheerful combination in the bedroom for girl
   **Target:** A bright and lively white-and-red color scheme in a girl's bedroom, creating a cheerful ambiance.
2. **Source:** vintage photograph of a young boy feeding pigeons .
   **Target:** A charming vintage photograph capturing a young boy feeding a flock of pigeons in a bustling city square.
3. **Source:** businessman with smartphone sitting on ledge by the sea
   **Target:** Serene coastal view as a businessman sits on a ledge by the sea, using his smartphone.
4. **Source:** a tourist taking a photograph of river looking west towards suspension bridge and office
   **Target:** Tourist snaps photo of suspension bridge and office building across the river.
5. **Source:** glass of foods and food product on a sunny day
   **Target:** An assortment of food items and products displayed in a glass container, illuminated by bright sunshine.
6. **Source:** turtles and large fish in the pond
   **Target:** A tranquil pond where large fish and turtles coexist peacefully, creating a harmonious natural habitat.
7. **Source:** the frescoes inside the dome
   **Target:** The elaborate and intricate paintings or artworks adorning the inner surface of the dome, typically found in religious buildings.

8. **Source:** fight over a loose ball
   **Target:** Intense competition as players struggle to gain control of a loose ball during the game.
9. **Source:** love this winter picture by person .
   **Target:** Mesmerizing winter landscape by person: serene snowy scenery with gentle snowflakes, skillfully framed with perfect contrast and depth.
10. **Source:** one of several paths through the woods .
    **Target:** A narrow forest path, one among many weaving through the lush trees, underbrush, and dappled sunlight.
11. **Source:** ripe strawberries falling through the water .
    **Target:** Juicy ripe strawberries plummeting through a stream of water, splashing and creating ripples in the liquid.
12. **Source:** a city reflected on a red sunglasses .
    **Target:** The cityscape reflected on a pair of red sunglasses, creating a distorted but fascinating view of the urban environment.
13. **Source:** man driving a car through the mountains
    **Target:** A man confidently navigating a winding mountain road with breathtaking views.
14. **Source:** maritime museum from the historical dock .
    **Target:** A museum dedicated to seafaring history, located on a historic dock where visitors can view a collection of artifacts and vessels.
15. **Source:** tree hollow and green leaves of a tree top in summer
    **Target:** Amidst lush green leaves on the top of a tree, a hollow creates a natural shelter, typical of summer foliage.
16. **Source:** musician of musical group performs on stage on the first day of festival
    **Target:** On the opening day of the festival, a musician from a musical group performs energetically on stage to a lively crowd.

**Bard:**

1. **Source:** man driving a car through the mountains
   **Target:** A man drives his car through the mountains, the road winding its way through the towering peaks.
2. **Source:** a bicycle hanging above the entrance to a store
   **Target:** A bicycle suspended from a chain above the entrance to a bike shop advertises their services.
3. **Source:** government agency released underwater footage of the unique movements of starfish
   **Target:** New footage released by government agency shows starfish swimming in unique ways, shedding light on their underwater behavior.
4. **Source:** unique red chair among other white chairs at the stadium
   **Target:** Red chair stands out among white seats at stadium, a beacon of color.
5. **Source:** person looks comfortable as he connects with a free - kick during the session and is in line to return against hull on saturday
   **Target:** The player strikes a free-kick with confidence during training, looking sharp and ready for his return to the team against Hull on Saturday.
6. **Source:** animal in front of a white background
   **Target:** An animal stands out against the stark white background, its feathers on full display.
7. **Source:** a mother and daughter lying on a lawn
   **Target:** A mother and daughter embrace in the grass, enjoying the warmth of the sun and the sound of birdsong.
8. **Source:** sign is seen outside the home
   **Target:** A for sale sign stands in the front yard, a reminder that change is always on the horizon.
9. **Source:** portrait of person against an abstract background stock photo
   **Target:** person's face stands out against a swirling, abstract background, their expression one of mystery and intrigue.
10. **Source:** state flag waving on an isolated white background .
    **Target:** The state flag waves proudly against a stark white background. The colors of the flag are vibrant and the design is intricate, representing the history and culture of the state
11. **Source:** actor wears a gorgeous blush pink coloured gown at festival .
    **Target:** Actor stuns in blush pink gown at festival, her beauty a match for the flowers in bloom.
12. **Source:** person answering the phones again at the office .
    **Target:** A friendly voice answers the phone, eager to help the caller with their needs.

13. **Source:** little boy sitting on the grass with drone and remote controller
    **Target:** A young boy's eyes light up as he launches his drone into the sky, his imagination taking flight with it.
14. **Source:** golfer competes during day held
    **Target:** Golfer focuses on the ball as she competes in a tournament on a beautiful day.
15. **Source:** golden fish in a bowl
    **Target:** A golden fish swims lazily in a bowl, its scales shimmering in the sunlight. The fish is a beautiful shade of orange, with black spots on its fins and tail.
16. **Source:** businessman with smartphone sitting on ledge by the sea
    **Target:** A businessman relaxes on a seaside ledge, checking his phone and enjoying the view.

**Human:**
1. **Source:** Honey buttermilk biscuits on a cooling rack being drizzled with honey
   **Target:** A warm stack of freshly baked honey buttermilk biscuits, sit on a cooling rack as they are drizzled with golden honey
2. **Source:** happy corgi time
   **Target:** Delighted corgi stands in the hallway, looking at its owner
3. **Source:** <PERSON> dog looking at dirt from the ground
   **Target:** <Person>'s dog, lying on the ground, looks at the dirt
4. **Source:** navy vintage pants - lime green bag - ivory Maison Simons t-shirt - Zara clogs
   **Target:** A young beautiful lady wearing navy vintage pants and ivory Maison Simons t-shirt, is holding a lime green bag.
5. **Source:** Ooak Barbie City Shine
   **Target:** A custom-made Barbie doll with a city-inspired look shines brightly
6. **Source:** Real Wedding on a NYC Rooftop
   **Target:** a couple is kissing each other during their rooftop wedding in NYC
7. **Source:** the proud of my beloved italian bracco after leg amputation due to a tumor.
   **Target:** my italian bracco lied down proudly under the sunshile, despite of leg amputation due to a tumor.
8. **Source:** Pineapple Wearing Headphones Art Print by Philip Haynes
   **Target:** An art from Philip Haynes depicts a pineapple that wears headphones
9. **Source:** Ominous thunderclouds behind the Capitol Building
   **Target:** Thunderclouds loom over the Capitol Building, casting a dark shadow
10. **Source:** Steampunk woman with gun
    **Target:** A fierce and stylish steampunk woman holds a toy revolver in her hands
11. **Source:** a new watch with some old friends
    **Target:** The watch sits besides a cartoon picture, evoking memories of cherished times shared with long-time friends
12. **Source:** Particularly important to Africa is the East African Highland Banana (EAHB), a staple food for 80 million people. Uganda alone has about 120 varieties of this type of banana.
    **Target:** An African man holds a bunch of bananas, which is particularly important to Africa
13. **Source:** Electric Blue Guitar There Goes My Hero, Rock The Vote, <PERSON>, <PERSON>, Music Photo, Red Eyes, Photo Quotes, Electric Blue, Music Lyrics
    **Target:** <PERSON> is playing an electric blue guitar, eyes bloodshot from the stage lights
14. **Source:** Advanced Bicycle Skills Video - Valuable Video for Safe Cycl
    **Target:** A Cyclist is demonstrating advanced bicycle skills in a video that will help people stay safe.
15. **Source:** grilled turkey pesto sandwich
    **Target:** A grilled turkey pesto sandwich with melted cheese and fresh arugula is served on a plate.
16. **Source:** Actress <PERSON> during the launch of international fashion brand Forever 21 store at a mall in Mumbai on Saturday, October 12th, 2013.
    **Target:** The young beautiful actress attended the launch of fashion brand Forever 21 at a mall.

**MSCOCO:**
For the meta-input-output sampling using the MSCOCO strategy, we utilize the fact that there are five different captions associated with each image. In our approach, we randomly select two texts from the available five, with one serving as the meta-input and the other as the meta-output. Below is a list of the captions we employ for this purpose.
1. **Caption 1 :** A herd of goats walking down a road way.
   **Caption 2 :** Three lambs stand next to each other and look different directions.

**Caption 3 :** The animals standing in the clearing are 3 varieties of sheep.
**Caption 4 :** Three small sheep are standing on a road.
**Caption 5 :** Some animals are standing on a dirt path

2. **Caption 1 :** A boy is preparing to toss a frisbie while another boy is sitting in the background in a park.
   **Caption 2 :** Several people are out in the woods on a path playing a game.
   **Caption 3 :** A man in a park playing a throwing game.
   **Caption 4 :** A group of people that are hanging out together.
   **Caption 5 :** A boy gets ready to throw a frisbee

3. **Caption 1 :** A pizza sitting on top of a metal pan.
   **Caption 2 :** The large pepperoni pizza is covered with chives.
   **Caption 3 :** A pizza that is sitting on a tray.
   **Caption 4 :** A large pizza with toppings sitting on a tray.
   **Caption 5 :** a pizza with fresh basil tomato sauce and cheese baked

4. **Caption 1 :** A woman sits on top of a motorcycle in a parade.
   **Caption 2 :** Woman wearing starts on helmet and shorts rides motorcycle
   **Caption 3 :** A woman wearing attire that matches her motorcycle is driving on.
   **Caption 4 :** A person that is on top of a motorcycle.
   **Caption 5 :** Woman on a motorcycle rides in a parade

5. **Caption 1 :** the people are sampling wine at a wine tasting.
   **Caption 2 :** Group of people tasting wine next to some barrels.
   **Caption 3 :** People are gathered around a man tasting wine.
   **Caption 4 :** A man pouring wine from casks for patrons
   **Caption 5 :** People gather around a table while sampling wine.

6. **Caption 1 :** A herd of sheep walking down a street in front of a bus.
   **Caption 2 :** There are three animals walking down the road.
   **Caption 3 :** a van is stuck behind a few traveling goats
   **Caption 4 :** a van that has some kind of animal out front of it
   **Caption 5 :** A herd of animals walking down the road behind a truck.

7. **Caption 1 :** A sandwich with meat and cheese sits on a plate with a small salad.
   **Caption 2 :** A sandwich with cheese and a bowl with a salad.
   **Caption 3 :** Two plates with sandwiches on them next to a bowl of vegetables.
   **Caption 4 :** A long sandwich and a salad is on a plate.
   **Caption 5 :** a sandwich and a bowl of vegetables on a plate

8. **Caption 1 :** A NASA airplane carrying a space shuttle on its back.
   **Caption 2 :** A large plan with a smaller plan on top of it.
   **Caption 3 :** A NASA airplane carrying the old Space Shuttle
   **Caption 4 :** A NASA airplane glides through the sky while carrying a shuttle.
   **Caption 5 :** This jet is carrying a space shuttle on it

9. **Caption 1 :** A one way sign under a blue street sign.
   **Caption 2 :** a view from below of a one way sign
   **Caption 3 :** A street sign stating that the road is one way beneath a blue sky.
   **Caption 4 :** A "One Way" street sign pointing to the right.
   **Caption 5 :** A one way road sign mounted above a street sign.

10. **Caption 1 :** A bowl of food containing broccoli and tomatoes.
    **Caption 2 :** A large salad is displayed in a silver metal bowl.
    **Caption 3 :** A bowl of food with tomatoes, sliced apples, and other greens
    **Caption 4 :** A silver bowl filled with various produce discards.
    **Caption 5 :** The salad in the bowl contains many fresh fruits and vegetables.

11. **Caption 1 :** a cake made to look like it has candy decorations on it
    **Caption 2 :** A photograph of a highly decorated cake on a table.
    **Caption 3 :** A cake decorated with lollipops and a piece of pie.
    **Caption 4 :** A piece of cake with lolypops, pie and caterpillar designs.
    **Caption 5 :** A layered cake with sweet treats and a caterpillar as decorations.

12. **Caption 1 :** A young man riding a skateboard on a cement walkway.
    **Caption 2 :** a guy riding a skateboard by a car
    **Caption 3 :** A young man on a skateboard near a car
    **Caption 4 :** an image of a boy on a skateboard doing tricks
    **Caption 5 :** A young man is riding on his skateboard.

Table A6: Performance comparison of LaCLIP trained with different meta-input-output strategies on CC12M.

(a) Zero-shot and Linear-probing Experiment Results

| Source | Food-101 | CIFAR-10 | CIFAR-100 | SUN397 | Cars | Aircraft | DTD | Pets | Caltech-101 | Flowers | STL-10 | EuroSAT | RESISC45 | GTSRB | Country211 | Average | ImageNet |
|---|---|---|---|---|---|---|---|---|---|---|---|---|---|---|---|---|---|
| Zero-shot | | | | | | | | | | | | | | | | | |
| ChatGPT | 57.0 | 71.1 | 38.9 | 51.2 | 31.6 | 3.9 | 25.5 | 63.0 | 80.8 | 36.9 | 92.9 | 24.5 | 39.6 | 10.1 | 6.9 | 42.3 | 44.5 |
| Bard | 55.2 | 70.1 | 39.4 | 51.7 | 31.5 | 4.6 | 25.2 | 63.3 | 80.6 | 34.5 | 92.5 | 20.7 | 39.6 | 10.1 | 7.2 | 41.7 | 44.8 |
| MSCOCO | 54.9 | 66.3 | 39.1 | 52.6 | 29.0 | 4.2 | 24.9 | 67.7 | 79.3 | 33.1 | 93.8 | 27.8 | 38.2 | 13.2 | 7.1 | 42.1 | 44.6 |
| Human | 56.4 | 69.1 | 39.1 | 51.7 | 31.4 | 3.8 | 22.9 | 68.1 | 80.6 | 38.4 | 94.3 | 26.9 | 43.0 | 11.7 | 7.5 | 43.0 | 45.1 |
| Linear-Probing | | | | | | | | | | | | | | | | | |
| ChatGPT | 81.5 | 94.0 | 79.4 | 73.0 | 77.2 | 54.7 | 75.1 | 87.1 | 92.2 | 96.0 | 97.3 | 96.6 | 92.3 | 81.0 | 19.9 | 79.8 | 71.2 |
| Bard | 82.0 | 93.7 | 79.4 | 72.7 | 77.6 | 53.8 | 74.4 | 86.3 | 92.0 | 95.7 | 97.1 | 96.2 | 92.5 | 81.7 | 19.6 | 79.6 | 71.2 |
| MSCOCO | 81.9 | 94.1 | 79.2 | 73.3 | 76.0 | 53.4 | 75.4 | 86.8 | 92.8 | 95.9 | 97.6 | 96.5 | 92.7 | 82.5 | 19.4 | 79.8 | 71.3 |
| Human | 82.3 | 94.2 | 79.4 | 73.3 | 76.2 | 55.1 | 75.6 | 87.0 | 92.0 | 96.3 | 97.5 | 96.2 | 92.8 | 81.3 | 19.8 | 79.9 | 71.3 |

(b) Few-shot Experiment Results

| Source | Food-101 | CIFAR-10 | CIFAR-100 | SUN397 | Cars | Aircraft | DTD | Pets | Caltech-101 | Flowers | STL-10 | EuroSAT | RESISC45 | GTSRB | Country211 |
|---|---|---|---|---|---|---|---|---|---|---|---|---|---|---|---|
| ChatGPT | 88.8±0.5 | 78.4±0.6 | 83.3±0.6 | 97.7±0.2 | 93.4±0.4 | 66.5±1.0 | 84.4±0.6 | 92.5±0.4 | 98.6±0.2 | 98.0±0.2 | 94.3±0.3 | 84.0±0.5 | 92.3±0.4 | 73.7±0.8 | 45.6±0.7 |
| Bard | 89.2±0.5 | 80.1±0.6 | 83.4±0.6 | 97.7±0.2 | 93.3±0.4 | 66.3±1.0 | 84.3±0.6 | 93.2±0.4 | 98.6±0.2 | 98.1±0.2 | 94.9±0.3 | 83.2±0.5 | 92.2±0.4 | 74.2±0.8 | 45.6±0.7 |
| MSCOCO | 88.6±0.5 | 79.5±0.6 | 82.7±0.6 | 97.8±0.2 | 93.7±0.4 | 65.1±1.0 | 84.4±0.6 | 92.5±0.4 | 98.7±0.2 | 98.1±0.2 | 95.0±0.3 | 84.9±0.5 | 91.6±0.4 | 74.3±0.8 | 44.9±0.7 |
| Human | 88.6±0.5 | 78.4±0.6 | 83.2±0.6 | 97.7±0.2 | 93.7±0.4 | 66.1±1.0 | 84.7±0.6 | 93.0±0.4 | 98.6±0.2 | 98.2±0.2 | 94.4±0.3 | 83.5±0.5 | 92.2±0.4 | 74.1±0.8 | 45.7±0.7 |

13. **Caption 1 :** A small brown dog sitting on display behind a window.
    **Caption 2 :** A small fuzzy dog stares longingly out a window.
    **Caption 3 :** The dog is brown shaggy with a red collar.
    **Caption 4 :** A dog sits alone and stares out of a window.
    **Caption 5 :** A furry and cute dog sitting in a window looking outside.
14. **Caption 1 :** A herd of sheep standing on a lush green hillside.
    **Caption 2 :** Several animals standing on the side of a hill.
    **Caption 3 :** A number of sheep eat on a steep grassy hill.
    **Caption 4 :** a couple of sheep are standing in some grass
    **Caption 5 :** The side of a small hill of grass with several sheep grazing in the grass and houses in the background on the upper hill.
15. **Caption 1 :** The tennis player on the blue court has his racquet raised.
    **Caption 2 :** A man swinging a tennis racket at a pro tennis match.
    **Caption 3 :** A tennis player wearing a NIKE shirt swings his racket
    **Caption 4 :** Man posing in front of the camera holding up a tennis racket.
    **Caption 5 :** A man wearing a white shirt playing tennis.
16. **Caption 1 :** A surfer riding a wave in a tempestuous ocean
    **Caption 2 :** Man in body suit surfing on a large wave.
    **Caption 3 :** A surfer is sideways on a wave of water on a surfboard.
    **Caption 4 :** The surfer is riding sideways along a wave.
    **Caption 5 :** a surfer wearing a wet suit is surfing on a white board

## C.2 Detailed Experiment Results on Meta-Input-Output

We present a detailed analysis of the experiment results comparing different meta-input-output strategies. Specifically, for each of the four meta-input-output strategy (*ChatGPT, Bard, Human, MSCOCO*), we use this specific strategy as example candidates for LLaMA ICL, and generate a rewrite for every text in CC12M. Then we train four LaCLIP models, each model trained with the original captions and the rewrite version of one specific meta-input-output strategy. The comprehensive results of these experiments are summarized in Table A6. The results indicate that different meta-input-output strategy achieves similar performance.

## D. Augmentation Strategy Details

To help understand the effect of our proposed language rewriting strategy by LLaMA ICL, here we compare our proposed strategy with two widely used language augmentation baselines: EDA [58] and back translation [50].

- **EDA** contains four types of different randomly performed augmentation operations: Synonym Replacement, Random Insertion, Random Swap, and Random Deletion. We used the official implementation and kept all the default parameters as used in [58].
- **Back Translation** first translates the text to another language and then translate it back to English to generate slightly different version of the text. We chose four different languages for our experiments: Spanish, French, German and Italic languages.

### D.1 Augmentation Qualitative Comparison

We begin by presenting qualitative comparisons of different text augmentation strategies. It is observed that the EDA and back translation approaches primarily focus on word-level modifications, often preserving the sentence structures and leading to limited diversity in the rewritten texts. In contrast, our LLM-based augmentation strategy demonstrates the ability to substantially alter the sentence structure while maintaining the original key concepts and meaning intact. This results in more effective, diverse, and enriched rewritten texts, highlighting the superiority of our approach over word-based methods.

- **Original:** Handmade mirror in a wooden toned frame texture cracked paint with reflection green apple on the table. Handmade mirror in a wooden toned frame texture cracked stock photos
  **EDA Aug 1:** handmade mirror in a wooden toned frame texture cracked on with reflection green apple paint the table handmade mirror in a wooden toned frame texture cracked stock photos
  **EDA Aug 2:** handmade mirror in a wooden toned frame texture crock up cracked paint with reflection green apple angstrom unit on the table handmade mirror in a wooden toned frame texture cracked stock photos
  **Back Translation Spanish:** Hand-made mirror in the toned wood frame cracked texture paint with green apple reflection on the table. Hand-made mirror in the toned wood frame cracked texture stock photos
  **Back Translation French:** Hand-made mirror in a wood toned frame texture cracked paint with green apple reflection on the table. Hand-made mirror in a wood toned frame texture cracked stock photos
  **Back Translation German:** Handmade mirror in a wooden toned frame texture cracked color with reflection green apple on the table. Handmade mirror in a wooden toned frame texture cracked stock photos
  **LLM ChatGPT:** Mirror on wooden painted frame, over cracked stone, with green apple on the table. Handmade mirror in a wooden toned frame texture cracked stock photo
  **LLM Bard:** Green apple, still life, studio photography, abstract background
  **LLM MSCOCO:** a mirror on a table
  **LLM Human:** A closeup on a cracked mirror that reflects an apple on the table. The frame is covered in a golden toned wood grain texture.
- **Original:** Traffic jam on the road, a lot of cars which go towards each other and to the different directions
  **EDA Aug 1:** traffic jam on the road a lot of cars which go towards each other and to the different focus
  **EDA Aug 2:** traffic jam on the road a lot of cars which go towards each other and dissimilar to the different directions
  **Back Translation Spanish:** Traffic jam on the road, a lot of cars going towards the other and to the different directions
  **Back Translation French:** Traffic jams on the road, lots of cars going towards each other and towards different directions
  **Back Translation German:** Traffic jam on the road, a lot of cars that go to each other and to the different directions
  **LLM ChatGPT:** Traffic jam on the road, with a lot of cars moving in different directions, as well as towards each other.
  **LLM Bard:** A lot of cars line the street, all heading toward the same intersection.

**LLM MSCOCO:** Traffic jam on the road with cars going in different directions, some cars are on the same lane, others are on different lanes.

**LLM Human:** A traffic jam on the road. There are a lot of cars and many of them are going in different directions.

- **Original:** Green young small newborn tree growing on concrete road among dry fallen leaves given concept of the life beginning or strong surviving

  **EDA Aug 1:** green young small newborn tree growing on concrete road among dry fallen leaves given concept of the life beginning or strong surviving

  **EDA Aug 2:** green young small new sprung tree growing on concrete road among dry fallen leaves given conception of the life beginning or strong surviving

  **Back Translation Spanish:** Young green small newborn tree growing on the concrete road between fallen dry leaves given the concept of the beginning of life or strong survival

  **Back Translation French:** Young young-born green tree growing on the concrete road among the dried leaves fallen according to the concept of the beginning of life or strong survivor

  **Back Translation German:** Green young little newborn tree growing on concrete road under dry fallen leaves given concept of life beginning or strong survival

  **LLM ChatGPT:** Bold forest green, from endless tangle of vines, create a strong vertical accent on the concrete road. On the right side, two brown fallen leaves, littering the road, sit quietly and gracefully, almost as though they are praying.

  **LLM Bard:** Green young small newborn tree growing on concrete road among dry fallen leaves giving concept of the life beginning or strong surviving

  **LLM MSCOCO:** A bare tree trunk surrounded by fallen leaves.

  **LLM Human:** a young and tall green tree on a concrete road among dry fallen leaves, its leaves on the floor. Concept is a strong surviving in harsh environment

- **Original:** Caused by the european solid wood floor lamp lamp creative decorative floor lamp floor lamp minimalist home japanese wood living room bedroom lights

  **EDA Aug 1:** room by the european solid wood floor lamp lamp creative decorative floor lamp floor lamp minimalist home japanese wood bedroom caused living lights

  **EDA Aug 2:** caused house by the european solid wood floor lamp lamp creative decorative floor lamp floor lamp minimalist home japanese wood living have room bedroom lights

  **Back Translation Spanish:** Caused by European solid wood floor lamp European decorative floor lamp creative minimalist floor lamp Japanese home wood living room bedroom lights

  **Back Translation French:** Cause of the European floor lamp in solid wood decorative floor lamp creative floor lamp minimalist floor lamp Japanese house living room light room bedroom

  **Back Translation German:** Conditioned by the European solid wood floor lamp lamp creative decorative floor lamp floor lamp minimalist house Japanese wood living room bedroom lights

  **LLM ChatGPT:** Due to the european style lighting system, the decorative floor lamp will enhance your living room decor.

  **LLM Bard:** Because of the european solid wood floor lamp, creative floor lamp, decorative floor lamp, floor lamp, minimalist home, japanese wood living room, bedroom lights.

  **LLM MSCOCO:** A furniture store with lamps on the shelves. One lamp is lit.

  **LLM Human:** The european solid wood floor lamp creative decorative floor lamp floor lamp minimalist home japanese wood living room bedroom lights

- **Original:** A man, likely a railroad employee, stands in the forefront of the station by the tracks. In the background, groups of people lean against the building, perhaps waiting for a train. In the foreground is the switch tower.

  **EDA Aug 1:** a man likely a railroad employee stands in the forefront of the station by the tracks in the background groups of inch people lean against the building perhaps waiting for a hulk train in the foreground is the transposition switch tower

  **EDA Aug 2:** a military personnel likely a railroad employee stands in the forefront of the station by the tracks in the background groups of people lean against the building perhaps ready and waiting for a train in the foreground is the throw tower

  **Back Translation Spanish:** A man, probably a railroad employee, is at the forefront of the station by the tracks. Deep down, groups of people lean on the building, perhaps waiting for a train. In the foreground is the switch tower.

  **Back Translation French:** A man, probably a railway employee, stands at the vanguard of the station by the tracks. In the background, groups of people lean against the building, perhaps waiting for a train.

  **Back Translation German:** A man, probably a railway worker, is standing at the top of the station

Table A7: Performance comparison of LaCLIP trained with different text augmentation strategies on CC12M.

(a) Zero-shot and Linear-probing Experiment Results

| Augmentation | Food-101 | CIFAR-10 | CIFAR-100 | SUN397 | Cars | Aircraft | DTD | Pets | Caltech-101 | Flowers | STL-10 | EuroSAT | RESISC45 | GTSRB | Country211 | Average | ImageNet |
|---|---|---|---|---|---|---|---|---|---|---|---|---|---|---|---|---|---|
| *Zero-shot* | | | | | | | | | | | | | | | | | |
| N/A (CLIP) | 50.8 | 64.9 | 38.5 | 44.7 | 24.1 | 2.4 | 19.4 | 64.1 | 77.4 | 33.2 | 91.0 | 20.1 | 38.9 | 7.3 | 5.1 | 38.8 | 40.2 |
| EDA [58] | 51.9 | 67.6 | 36.5 | 48.2 | 27.7 | 2.8 | 25.4 | 64.7 | 78.2 | 33.3 | 92.8 | 21.9 | 40.0 | 10.8 | 6.6 | 40.6 | 41.2 |
| Back Translation [50] | 49.3 | 71.0 | 36.7 | 47.9 | 27.8 | 3.7 | 25.7 | 63.9 | 77.4 | 32.0 | 90.6 | 22.0 | 41.3 | 10.7 | 6.1 | 40.4 | 41.6 |
| LLM (Ours) | **60.7** | **75.1** | **43.9** | **57.0** | **36.3** | **5.6** | **31.0** | **72.4** | **83.3** | **39.9** | **95.1** | **27.3** | **44.3** | **12.7** | **8.9** | **46.2** | **48.4** |
| *Linear-Probing* | | | | | | | | | | | | | | | | | |
| N/A (CLIP) | 81.6 | 93.8 | 79.3 | 72.0 | 75.1 | 52.6 | 75.6 | 86.2 | 92.2 | 95.3 | 97.3 | 96.7 | 93.1 | 80.6 | 19.7 | 79.4 | 70.3 |
| EDA [58] | 81.6 | 94.0 | 78.2 | 72.9 | 76.2 | 53.7 | 74.8 | 85.6 | 92.2 | 95.5 | 97.2 | 96.8 | 92.9 | 79.9 | **20.1** | 79.4 | 70.5 |
| Back Translation [50] | 81.8 | 94.2 | 78.2 | 73.0 | 77.5 | **54.6** | 75.5 | 87.1 | 91.6 | 96.0 | 97.5 | **97.1** | 93.1 | 80.0 | 20.0 | 79.8 | 70.7 |
| LLM (Ours) | **82.9** | **94.7** | **79.7** | **73.8** | **79.9** | 54.5 | **75.7** | **87.7** | **93.0** | **96.4** | **98.0** | 96.4 | 93.0 | **81.9** | 19.7 | **80.5** | **72.3** |

(b) Few-shot Experiment Results

| Augmentation | Food-101 | CIFAR-10 | CIFAR-100 | SUN397 | Cars | Aircraft | DTD | Pets | Caltech-101 | Flowers | STL-10 | EuroSAT | RESISC45 | GTSRB | Country211 |
|---|---|---|---|---|---|---|---|---|---|---|---|---|---|---|---|
| N/A (CLIP) | 87.0±0.5 | 77.5±0.6 | 82.1±0.7 | 97.2±0.2 | 90.9±0.5 | 62.0±1.0 | 83.3±0.6 | 91.1±0.5 | 98.2±0.2 | 97.6±0.2 | 92.6±0.4 | 83.4±0.5 | 91.2±0.4 | 70.6±0.8 | 44.3±0.7 |
| EDA [58] | 88.1±0.5 | 76.1±0.6 | 81.3±0.7 | 97.6±0.2 | 91.7±0.5 | 62.9±1.0 | 83.4±0.6 | 91.9±0.5 | 98.4±0.2 | 97.8±0.2 | 93.5±0.3 | **84.3±0.5** | 91.6±0.4 | 68.4±0.8 | 44.6±0.7 |
| Back Trans [50] | 88.1±0.5 | 76.9±0.6 | 82.5±0.7 | 97.5±0.2 | 91.8±0.4 | 65.1±1.0 | 83.7±0.6 | 92.5±0.4 | 98.3±0.2 | 97.9±0.2 | 94.2±0.3 | 83.3±0.5 | 91.1±0.4 | 70.8±0.8 | 45.1±0.7 |
| LLM (Ours) | **89.9±0.5** | **81.3±0.5** | **85.0±0.6** | **98.0±0.2** | **95.3±0.3** | **68.1±1.0** | **84.9±0.6** | **93.4±0.4** | **98.9±0.2** | **98.4±0.2** | **95.9±0.2** | 83.0±0.5 | **92.4±0.4** | **76.4±0.8** | **46.7±0.7** |

on the tracks. In the background, groups of people are leaning against the building, perhaps waiting for a train.
**LLM ChatGPT:** A man, likely a railroad employee, stands by the tracks in front of the station. In the background, groups of people lean against the building. In the foreground is the switch tower.
**LLM Bard:** man leaning against the rail station and switch tower in a dark night with a fog
**LLM MSCOCO:** A portrait of the man in the front of the station is seen in the upper left. In the lower right is a man leaning on a post with his arms crossed.
**LLM Human:** An image of a man, likely a railroad employee, standing in the foreground of a train station by the tracks. In the background are groups of people, some leaning against the building, which could be waiting for a train. In the foreground are the tracks with a switch tower in the distance.

## D.2 Detailed Experiment Results on Augmentation Strategy

We conducted a quantitative comparison of different augmentation strategies while ensuring a fair evaluation by generating a consistent number of augmented texts per original sentence (i.e., 4).

For the EDA strategy, we created 4 distinct versions of each sentence by randomly applying their predefined augmentation operations. As for the back translation approach, we translated the original texts into four different languages (Spanish, French, German, and Italic languages) and then back to English, resulting in 4 rewritten versions of the original texts. In our LLM-based augmentation, we used LLaMA ICL to generate 4 augmentations prompted by the 4 predefined meta-input-output pairs (ChatGPT, Bard, Human, and MSCOCO).

A comprehensive comparison of these strategies is presented in Table A7. The results demonstrate that while the baseline augmentation strategies improve the performance of the vanilla CLIP baseline, our proposed LLM-based augmentation strategy consistently achieves superior results across various datasets and evaluation metrics, outperforming the other augmentation methods significantly.

Table A8: Performance comparison of CLIP and LaCLIP trained with different text augmentation strategies with different number of augmentations per original text on CC12M.

(a) Zero-shot and Linear-probing Experiment Results

| Augment / Num | Food-101 | CIFAR-10 | CIFAR-100 | SUN397 | Cars | Aircraft | DTD | Pets | Caltech-101 | Flowers | STL-10 | EuroSAT | RESISC45 | GTSRB | Country211 | Average | ImageNet |
|---|---|---|---|---|---|---|---|---|---|---|---|---|---|---|---|---|---|
| Zero-shot | | | | | | | | | | | | | | | | | |
| N/A (CLIP) / 0 | 50.8 | 64.9 | 38.5 | 44.7 | 24.1 | 2.4 | 19.4 | 64.1 | 77.4 | 33.2 | 91.0 | 20.1 | 38.9 | 7.3 | 5.1 | 38.8 | 40.2 |
| EDA / 1 | 52.2 | 66.2 | 34.3 | 46.6 | 25.6 | 3.6 | 22.2 | 64.5 | 79.3 | 33.5 | 90.9 | 24.1 | 37.6 | 13.4 | 5.6 | 40.0 | 40.2 |
| EDA / 2 | 49.8 | 62.4 | 32.1 | 47.1 | 28.1 | 2.2 | 25.3 | 64.6 | 79.1 | 31.4 | 92.3 | 12.6 | 38.0 | 13.1 | 5.7 | 38.9 | 41.1 |
| EDA / 3 | 50.4 | 62.8 | 35.4 | 49.7 | 26.8 | 2.5 | 24.5 | 69.5 | 77.4 | 33.1 | 92.8 | 24.9 | 37.3 | 15.1 | 6.7 | 40.6 | 41.7 |
| EDA / 4 | 51.9 | 67.6 | 36.5 | 48.2 | 27.7 | 2.8 | 25.4 | 64.7 | 78.2 | 33.3 | 92.8 | 21.9 | 40.0 | 10.8 | 6.6 | 40.6 | 41.2 |
| Back Trans / 1 | 49.7 | 61.5 | 34.6 | 45.5 | 26.7 | 4.0 | 20.7 | 59.2 | 77.2 | 32.1 | 88.2 | 27.1 | 40.0 | 12.6 | 5.8 | 39.0 | 40.1 |
| Back Trans / 2 | 50.0 | 55.4 | 35.5 | 44.3 | 29.0 | 5.2 | 21.0 | 67.4 | 78.5 | 32.6 | 89.4 | 19.6 | 38.4 | 7.6 | 6.2 | 38.7 | 41.0 |
| Back Trans / 3 | 49.9 | 67.3 | 37.6 | 46.9 | 26.7 | 4.1 | 22.8 | 65.7 | 76.8 | 34.3 | 91.7 | 20.0 | 34.3 | 12.5 | 6.3 | 39.8 | 41.5 |
| Back Trans / 4 | 49.3 | 71.0 | 36.7 | 47.9 | 27.8 | 3.7 | 25.7 | 63.9 | 77.4 | 32.0 | 90.6 | 22.0 | 41.3 | 10.7 | 6.1 | 40.4 | 41.6 |
| LLM (Ours) / 1 | 57.0 | 71.1 | 38.9 | 51.2 | 31.6 | 3.9 | 25.5 | 63.0 | 80.8 | 36.9 | 92.9 | 24.5 | 39.6 | 10.1 | 6.9 | 42.3 | 44.5 |
| LLM (Ours) / 2 | 57.0 | 70.3 | 41.3 | 54.2 | 34.2 | 5.8 | 29.0 | 64.0 | 79.5 | 38.5 | 94.4 | 33.0 | 38.6 | 9.1 | 8.2 | 43.8 | 46.5 |
| LLM (Ours) / 3 | 59.7 | 75.0 | 42.6 | 56.5 | 34.0 | 5.1 | 29.4 | 65.8 | 81.3 | 38.2 | 94.7 | 18.7 | 42.4 | 13.4 | 8.7 | 44.4 | 47.7 |
| LLM (Ours) / 4 | **60.7** | **75.1** | **43.9** | **57.0** | **36.3** | **5.6** | **31.0** | **72.4** | **83.3** | **39.9** | **95.1** | 27.3 | **44.3** | 12.7 | **8.9** | **46.2** | **48.4** |
| Linear-Probing | | | | | | | | | | | | | | | | | |
| N/A (CLIP) /0 | 81.6 | 93.8 | 79.3 | 72.0 | 75.1 | 52.6 | 75.6 | 86.2 | 92.2 | 95.3 | 97.3 | 96.7 | 93.1 | 80.6 | 19.7 | 79.4 | 70.3 |
| EDA / 1 | 81.5 | 93.3 | 78.0 | 72.1 | 75.6 | 53.1 | 76.5 | 85.9 | 91.5 | 95.8 | 97.3 | 96.4 | 92.6 | 80.0 | 19.9 | 79.3 | 70.4 |
| EDA / 2 | 81.4 | 94.1 | 80.2 | 72.5 | 76.7 | 52.9 | 75.7 | 85.8 | 92.1 | 95.7 | 97.2 | 96.7 | 92.7 | 81.6 | 19.9 | 79.7 | 70.6 |
| EDA / 3 | 81.3 | 93.6 | 78.8 | 72.3 | 74.5 | 53.3 | 75.1 | 86.0 | 91.1 | 95.6 | 97.3 | 96.7 | 93.0 | 79.1 | 19.7 | 79.2 | 70.6 |
| EDA / 4 | 81.6 | 94.0 | 78.2 | 72.9 | 76.2 | 53.7 | 74.8 | 85.6 | 92.2 | 95.5 | 97.2 | 96.8 | 92.9 | 79.9 | 20.1 | 79.4 | 70.5 |
| Back Trans / 1 | 81.5 | 93.4 | 78.3 | 72.4 | 76.9 | 52.5 | 74.8 | 85.7 | 92.0 | 95.5 | 97.4 | 96.9 | **93.2** | 81.6 | 19.8 | 79.5 | 70.5 |
| Back Trans / 2 | 81.5 | 93.9 | 78.5 | 72.4 | 76.3 | 52.8 | 74.5 | 86.2 | 91.7 | 95.5 | 97.5 | 96.8 | 92.4 | 80.5 | 19.4 | 79.3 | 70.5 |
| Back Trans / 3 | 81.6 | 93.5 | 78.0 | 72.4 | 75.9 | 52.1 | 73.8 | 86.2 | 92.1 | 95.1 | 97.3 | 96.5 | 92.3 | 79.4 | 19.9 | 79.1 | 70.5 |
| Back Trans / 4 | 81.8 | 94.2 | 78.2 | 73.0 | 77.5 | 54.6 | 75.5 | 87.1 | 91.6 | 96.0 | 97.5 | **97.1** | 93.1 | 80.0 | 20.0 | 79.8 | 70.7 |
| LLM (Ours) / 1 | 81.8 | 94.3 | 79.7 | 73.3 | 77.5 | 55.0 | 75.4 | 87.4 | 92.5 | 96.3 | 97.6 | 96.9 | 92.6 | 81.3 | **20.2** | 80.1 | 71.2 |
| LLM (Ours) / 2 | 82.3 | 94.0 | 79.1 | 73.3 | 77.6 | 52.7 | 76.0 | 86.8 | 91.8 | 96.1 | 97.7 | 96.6 | 93.1 | 83.3 | 20.1 | 80.0 | 71.7 |
| LLM (Ours) / 3 | 82.3 | 94.7 | 80.0 | 73.7 | 79.2 | **56.0** | 75.7 | 87.0 | 92.9 | 96.2 | 98.0 | 96.6 | 92.9 | 83.1 | 20.0 | 80.6 | 71.9 |
| LLM (Ours) / 4 | **82.9** | **94.7** | **79.7** | **73.8** | **79.9** | 54.5 | **75.7** | **87.7** | **93.0** | **96.4** | **98.0** | 96.4 | 93.0 | **81.9** | 19.7 | **80.5** | **72.3** |

(b) Few-shot Experiment Results

| Augment / Num | Food-101 | CIFAR-10 | CIFAR-100 | SUN397 | Cars | Aircraft | DTD | Pets | Caltech-101 | Flowers | STL-10 | EuroSAT | RESISC45 | GTSRB | Country211 |
|---|---|---|---|---|---|---|---|---|---|---|---|---|---|---|---|
| N/A (CLIP) / 0 | 87.0$_{\pm0.5}$ | 77.5$_{\pm0.6}$ | 82.1$_{\pm0.7}$ | 97.2$_{\pm0.2}$ | 90.9$_{\pm0.5}$ | 62.0$_{\pm1.0}$ | 83.3$_{\pm0.6}$ | 91.1$_{\pm0.4}$ | 98.2$_{\pm0.2}$ | 97.6$_{\pm0.2}$ | 92.6$_{\pm0.4}$ | 83.4$_{\pm0.5}$ | 91.2$_{\pm0.4}$ | 70.6$_{\pm0.8}$ | 44.3$_{\pm0.7}$ |
| EDA / 1 | 87.6$_{\pm0.5}$ | 75.4$_{\pm0.6}$ | 81.3$_{\pm0.7}$ | 97.4$_{\pm0.2}$ | 91.3$_{\pm0.5}$ | 62.6$_{\pm1.0}$ | 83.5$_{\pm0.6}$ | 91.5$_{\pm0.5}$ | 98.2$_{\pm0.2}$ | 97.8$_{\pm0.2}$ | 93.0$_{\pm0.3}$ | 83.2$_{\pm0.5}$ | 91.4$_{\pm0.4}$ | 68.9$_{\pm0.8}$ | 44.4$_{\pm0.7}$ |
| EDA / 2 | 87.9$_{\pm0.5}$ | 77.3$_{\pm0.6}$ | 82.0$_{\pm0.6}$ | 97.4$_{\pm0.2}$ | 91.9$_{\pm0.4}$ | 62.8$_{\pm1.0}$ | 83.5$_{\pm0.6}$ | 92.1$_{\pm0.4}$ | 98.4$_{\pm0.2}$ | 97.8$_{\pm0.2}$ | 93.6$_{\pm0.3}$ | 82.8$_{\pm0.6}$ | 91.6$_{\pm0.4}$ | 70.0$_{\pm0.8}$ | 44.8$_{\pm0.7}$ |
| EDA / 3 | 87.5$_{\pm0.5}$ | 76.5$_{\pm0.6}$ | 82.0$_{\pm0.7}$ | 97.6$_{\pm0.2}$ | 91.2$_{\pm0.5}$ | 62.7$_{\pm1.0}$ | 83.8$_{\pm0.6}$ | 91.3$_{\pm0.5}$ | 98.2$_{\pm0.2}$ | 97.7$_{\pm0.2}$ | 94.2$_{\pm0.3}$ | 84.0$_{\pm0.5}$ | 91.4$_{\pm0.4}$ | 72.0$_{\pm0.8}$ | 44.3$_{\pm0.7}$ |
| EDA / 4 | 88.1$_{\pm0.5}$ | 76.1$_{\pm0.6}$ | 81.3$_{\pm0.7}$ | 97.6$_{\pm0.2}$ | 91.7$_{\pm0.5}$ | 62.9$_{\pm1.0}$ | 83.4$_{\pm0.6}$ | 91.9$_{\pm0.6}$ | 98.4$_{\pm0.2}$ | 97.8$_{\pm0.2}$ | 93.5$_{\pm0.3}$ | **84.3**$_{\pm0.5}$ | 91.6$_{\pm0.4}$ | 68.4$_{\pm0.8}$ | 44.6$_{\pm0.7}$ |
| Back Trans / 1 | 87.8$_{\pm0.5}$ | 76.4$_{\pm0.6}$ | 81.8$_{\pm0.7}$ | 97.4$_{\pm0.2}$ | 91.7$_{\pm0.5}$ | 63.4$_{\pm1.0}$ | 83.8$_{\pm0.6}$ | 91.7$_{\pm0.5}$ | 98.3$_{\pm0.2}$ | 97.7$_{\pm0.2}$ | 93.1$_{\pm0.3}$ | 83.9$_{\pm0.5}$ | 91.6$_{\pm0.4}$ | 70.1$_{\pm0.8}$ | 44.7$_{\pm0.8}$ |
| Back Trans / 2 | 87.8$_{\pm0.5}$ | 75.6$_{\pm0.6}$ | 81.6$_{\pm0.7}$ | 97.5$_{\pm0.2}$ | 92.3$_{\pm0.4}$ | 62.8$_{\pm1.0}$ | 83.7$_{\pm0.6}$ | 92.5$_{\pm0.4}$ | 98.3$_{\pm0.2}$ | 97.8$_{\pm0.2}$ | 93.6$_{\pm0.3}$ | 83.8$_{\pm0.5}$ | 91.1$_{\pm0.4}$ | 68.8$_{\pm0.8}$ | 44.7$_{\pm0.7}$ |
| Back Trans / 3 | 88.2$_{\pm0.5}$ | 77.0$_{\pm0.6}$ | 82.8$_{\pm0.6}$ | 97.4$_{\pm0.2}$ | 91.7$_{\pm0.4}$ | 62.6$_{\pm1.0}$ | 83.8$_{\pm0.6}$ | 91.6$_{\pm0.5}$ | 98.3$_{\pm0.2}$ | 97.7$_{\pm0.2}$ | 93.3$_{\pm0.3}$ | 83.1$_{\pm0.5}$ | 91.8$_{\pm0.4}$ | 71.0$_{\pm0.8}$ | 45.0$_{\pm0.7}$ |
| Back Trans / 4 | 88.1$_{\pm0.5}$ | 76.9$_{\pm0.6}$ | 82.5$_{\pm0.7}$ | 97.5$_{\pm0.2}$ | 91.8$_{\pm0.4}$ | 65.1$_{\pm1.0}$ | 83.7$_{\pm0.6}$ | 92.5$_{\pm0.4}$ | 98.3$_{\pm0.2}$ | 97.9$_{\pm0.2}$ | 94.2$_{\pm0.3}$ | 83.3$_{\pm0.5}$ | 91.1$_{\pm0.4}$ | 70.8$_{\pm0.8}$ | 45.1$_{\pm0.7}$ |
| LLM (Ours) / 1 | 88.8$_{\pm0.5}$ | 78.4$_{\pm0.6}$ | 83.3$_{\pm0.6}$ | 97.7$_{\pm0.2}$ | 93.4$_{\pm0.4}$ | 66.5$_{\pm1.0}$ | 84.4$_{\pm0.6}$ | 92.5$_{\pm0.4}$ | 98.6$_{\pm0.2}$ | 98.0$_{\pm0.2}$ | 94.3$_{\pm0.3}$ | 84.0$_{\pm0.5}$ | 92.3$_{\pm0.4}$ | 73.7$_{\pm0.8}$ | 45.6$_{\pm0.7}$ |
| LLM (Ours) / 2 | 89.2$_{\pm0.5}$ | 79.1$_{\pm0.6}$ | 83.6$_{\pm0.6}$ | 97.9$_{\pm0.2}$ | 94.2$_{\pm0.4}$ | 65.6$_{\pm1.0}$ | 84.2$_{\pm0.6}$ | 93.2$_{\pm0.4}$ | 98.8$_{\pm0.2}$ | 98.2$_{\pm0.2}$ | 95.3$_{\pm0.3}$ | 83.6$_{\pm0.5}$ | 91.7$_{\pm0.4}$ | 75.6$_{\pm0.8}$ | 46.1$_{\pm0.7}$ |
| LLM (Ours) / 3 | 89.8$_{\pm0.5}$ | **82.5**$_{\pm0.5}$ | 84.2$_{\pm0.6}$ | **98.0**$_{\pm0.2}$ | 94.4$_{\pm0.4}$ | **68.5**$_{\pm1.0}$ | **85.0**$_{\pm0.6}$ | 93.4$_{\pm0.4}$ | 98.7$_{\pm0.2}$ | 98.4$_{\pm0.2}$ | **95.9**$_{\pm0.2}$ | **83.9**$_{\pm0.5}$ | 91.6$_{\pm0.4}$ | 75.1$_{\pm0.8}$ | **46.9**$_{\pm0.7}$ |
| LLM (Ours) / 4 | **89.9**$_{\pm0.5}$ | 81.3$_{\pm0.5}$ | **85.0**$_{\pm0.6}$ | **98.0**$_{\pm0.2}$ | **95.3**$_{\pm0.3}$ | 68.1$_{\pm1.0}$ | 84.9$_{\pm0.6}$ | **93.4**$_{\pm0.4}$ | **98.9**$_{\pm0.2}$ | **98.4**$_{\pm0.2}$ | **95.9**$_{\pm0.2}$ | 83.0$_{\pm0.5}$ | **92.4**$_{\pm0.4}$ | **76.4**$_{\pm0.8}$ | 46.7$_{\pm0.7}$ |

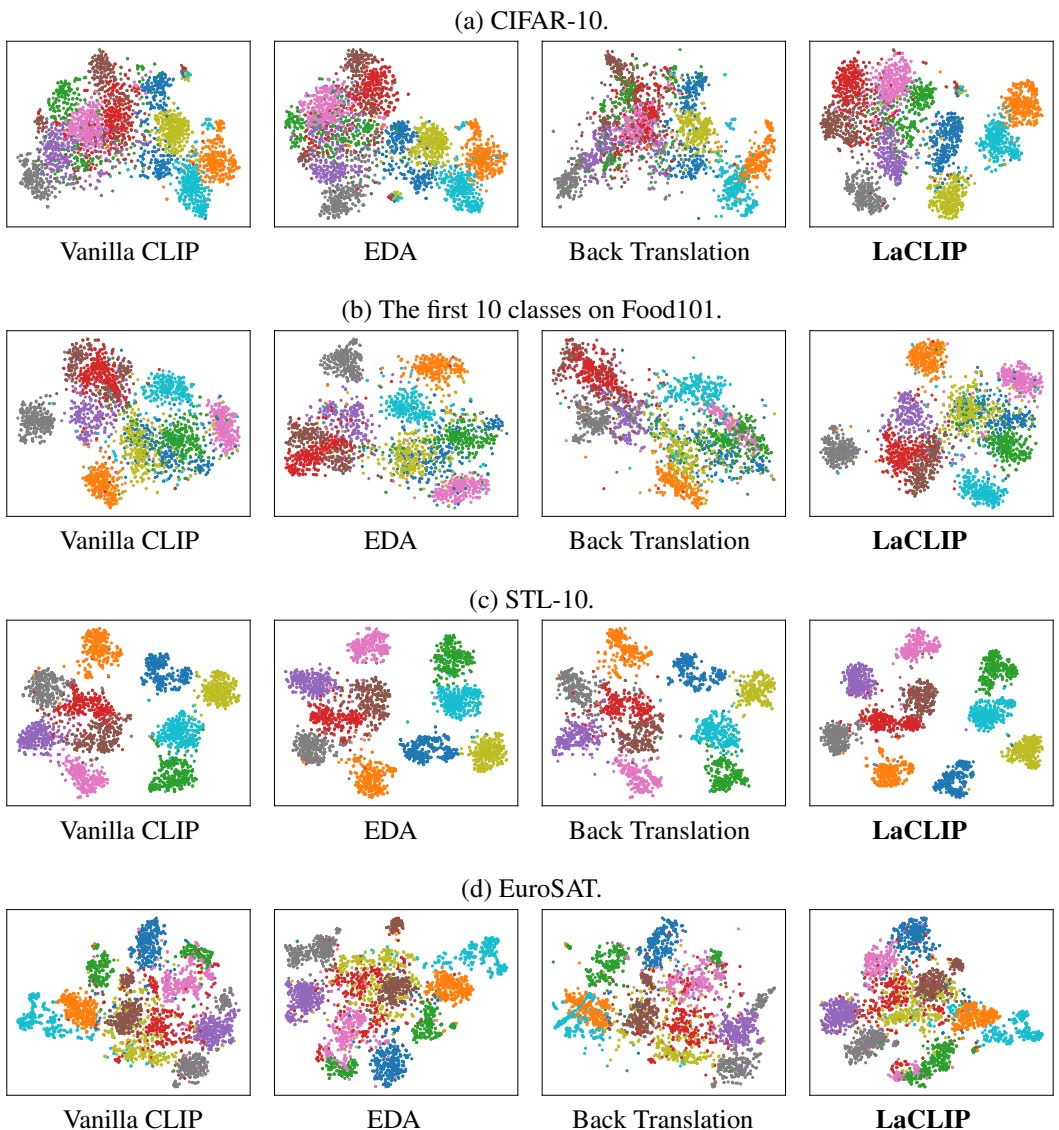

Figure A1: t-SNE visualization of image features learned from Vanilla CLIP, two baseline text augmentations strategies (EDA and back translation), and our proposed LaCLIP on CIFAR-10, Food101, STL-10, and EuroSAT datasets. Image features learned from our proposed LaCLIP have a clearer class boundaries and cluster centroids.

## E. Number of Augmentations per Original Text

We conducted experiments to investigate how the performance varies with the number of augmentations generated for each text and the differences between augmentation strategies as the number of augmentations per original text increases. We examined the performance of each strategy with 0 to 4 augmentations per original text, where 0 corresponds to vanilla CLIP without any text augmentation. Specifically, for each specific number of augmentations $k$: For EDA, we selected k versions out of the 4 generated versions. In the case of back translation, we used *Spanish, Spanish+French, Spanish+French+German*, and *Spanish+French+German+Italic languages* for $k = 1, 2, 3, 4$, respectively. Regarding our LLM-based augmentation, we used *ChatGPT, ChatGPT+Bard, Chat-GPT+Bard+MSCOCO*, and *ChatGPT+Bard+MSCOCO+Human* as augmentations corresponding to $k = 1, 2, 3, 4$, respectively.

The detailed comparison can be found in Table A8. From the results, we observe that the performance of the baseline augmentation strategies does not scale well with the number of augmentations per sentence, indicating limited diversity in the rewritten texts. This aligns with the findings in [58], where the best results are obtained with four different augmentations. In contrast, LaCLIP trained

Table A9: Performance comparison of CLIP, LaCLIP and LaCLIP-MT trained on CC12M and RedCaps.
(a) Zero-shot and Linear-probing Experiment Results

| Data | Model | Food-101 | CIFAR-10 | CIFAR-100 | SUN397 | Cars | Aircraft | DTD | Pets | Caltech-101 | Flowers | STL-10 | EuroSAT | RESISC45 | GTSRB | Country211 | Average | ImageNet |
|---|---|---|---|---|---|---|---|---|---|---|---|---|---|---|---|---|---|---|
| | | | | | | | | Zero-shot | | | | | | | | | | |
| CC12M | CLIP | 50.8 | 64.9 | 38.5 | 44.7 | 24.1 | 2.4 | 19.4 | 64.1 | 77.4 | 33.2 | 91.0 | 20.1 | 38.9 | 7.3 | 5.1 | 38.8 | 40.2 |
| | LaCLIP | 60.7 | 75.1 | 43.9 | 57.0 | 36.3 | 5.6 | 31.0 | 72.4 | 83.3 | 39.9 | 95.1 | 27.3 | 44.3 | 12.7 | 8.9 | 46.2 | 48.4 |
| | LaCLIP-MT | 59.2 | 69.5 | 39.0 | 56.8 | 34.4 | 5.5 | 30.7 | 72.8 | 83.1 | 42.5 | 95.2 | 24.8 | 43.4 | 13.1 | 8.3 | 45.2 | 49.0 |
| RedCaps | CLIP | 81.5 | 70.4 | 39.9 | 33.2 | 19.2 | 1.9 | 19.7 | 82.7 | 72.8 | 53.9 | 92.8 | 23.3 | 33.6 | 8.3 | 6.2 | 42.6 | 42.9 |
| | LaCLIP | 85.0 | 74.8 | 40.7 | 40.3 | 21.3 | 2.2 | 23.9 | 78.2 | 76.4 | 59.0 | 91.4 | 27.1 | 41.3 | 5.6 | 7.6 | 45.0 | 46.2 |
| | LaCLIP-MT | 84.2 | 74.9 | 43.1 | 40.5 | 23.0 | 1.9 | 24.0 | 84.7 | 77.1 | 60.9 | 91.0 | 31.9 | 40.3 | 6.1 | 7.9 | 46.1 | 48.1 |
| | | | | | | | | Linear-Probing | | | | | | | | | | |
| CC12M | CLIP | 81.6 | 93.8 | 79.3 | 72.0 | 75.1 | 52.6 | 75.6 | 86.2 | 92.2 | 95.3 | 97.3 | 96.7 | 93.1 | 80.6 | 19.7 | 79.4 | 70.3 |
| | LaCLIP | 82.9 | 94.7 | 79.7 | 73.8 | 79.9 | 54.5 | 75.7 | 87.7 | 93.0 | 96.4 | 98.0 | 96.4 | 93.0 | 81.9 | 19.7 | 80.5 | 72.3 |
| | LaCLIP-MT | 82.9 | 94.5 | 79.7 | 73.7 | 79.4 | 55.0 | 76.0 | 87.9 | 93.0 | 96.4 | 97.6 | 96.2 | 93.1 | 82.7 | 20.2 | 80.6 | 72.4 |
| RedCaps | CLIP | 89.1 | 94.1 | 78.8 | 65.6 | 74.0 | 52.5 | 73.2 | 91.5 | 91.4 | 97.7 | 98.0 | 96.3 | 93.5 | 80.8 | 17.0 | 79.6 | 71.8 |
| | LaCLIP | 90.1 | 94.3 | 78.5 | 66.6 | 77.6 | 53.6 | 73.9 | 90.8 | 91.5 | 97.9 | 97.6 | 96.6 | 92.7 | 80.8 | 17.2 | 80.0 | 71.9 |
| | LaCLIP-MT | 90.2 | 94.0 | 79.0 | 67.3 | 79.2 | 53.2 | 75.3 | 91.7 | 91.0 | 98.3 | 98.1 | 96.9 | 93.0 | 80.6 | 17.2 | 80.3 | 72.4 |

(b) Few-shot Experiment Results

| Model | Food-101 | CIFAR-10 | CIFAR-100 | SUN397 | Cars | Aircraft | DTD | Pets | Caltech-101 | Flowers | STL-10 | EuroSAT | RESISC45 | GTSRB | Country211 |
|---|---|---|---|---|---|---|---|---|---|---|---|---|---|---|---|
| | | | | | | Pre-trained on CC12M | | | | | | | | | |
| CLIP | 87.0±0.5 | 77.5±0.6 | 82.1±0.7 | 97.2±0.2 | 90.9±0.5 | 62.0±1.0 | 83.3±0.6 | 91.1±0.5 | 98.2±0.2 | 97.6±0.2 | 92.6±0.4 | 83.4±0.5 | 91.2±0.4 | 70.6±0.8 | 44.3±0.7 |
| LaCLIP | 89.9±0.5 | 81.3±0.5 | 85.0±0.6 | 98.0±0.2 | 95.3±0.3 | 68.1±1.0 | 84.9±0.6 | 93.4±0.4 | 98.9±0.2 | 98.4±0.2 | 95.9±0.2 | 83.0±0.5 | 92.4±0.4 | 76.4±0.8 | 46.7±0.7 |
| LaCLIP-MT | 89.5±0.5 | 80.1±0.5 | 84.4±0.6 | 98.0±0.2 | 94.8±0.4 | 69.6±1.0 | 84.6±0.6 | 93.7±0.4 | 98.8±0.2 | 98.4±0.2 | 96.0±0.2 | 83.8±0.5 | 92.0±0.4 | 76.8±0.7 | 46.4±0.7 |
| | | | | | | Pre-trained on RedCaps | | | | | | | | | |
| CLIP | 94.4±0.3 | 80.6±0.5 | 85.3±0.6 | 95.9±0.3 | 88.5±0.6 | 54.5±0.9 | 82.6±0.6 | 94.5±0.4 | 97.8±0.2 | 99.0±0.1 | 94.8±0.3 | 84.9±0.5 | 91.3±0.4 | 75.3±0.8 | 40.6±0.7 |
| LaCLIP | 95.8±0.3 | 81.4±0.5 | 85.4±0.6 | 96.2±0.3 | 90.9±0.5 | 58.8±1.0 | 82.4±0.6 | 94.1±0.4 | 98.0±0.2 | 99.2±0.1 | 95.6±0.2 | 86.2±0.5 | 92.1±0.4 | 76.5±0.8 | 42.6±0.7 |
| LaCLIP-MT | 95.9±0.3 | 81.8±0.5 | 86.0±0.6 | 96.5±0.3 | 91.4±0.5 | 58.1±1.0 | 82.7±0.6 | 94.8±0.4 | 98.2±0.2 | 99.3±0.1 | 95.4±0.2 | 87.5±0.4 | 92.2±0.4 | 76.5±0.8 | 42.5±0.7 |

with our LLM-based augmentation demonstrates good scalability with the number of augmentations. This can be attributed to the rich and diverse nature of LLaMA ICL in the rewriting process, allowing for continued performance improvement with more augmentations.

## F. t-SNE Visualizations

To gain a deeper understanding of the distinctions between the features learned from LaCLIP and vanilla CLIP, as well as the impact of different augmentation strategies used in LaCLIP training, we visualize the vision encoder features on different downstream datasets using t-SNE [57] in Figure A1. We generate feature visualizations for CIFAR-10, Food101, STL-10, and EuroSAT datasets, as they provide sufficient samples per class for meaningful visualizations. Other datasets have a limited number of samples per class in the test set, making it difficult to generate reliable visualizations. For Food101 we visualize the features from the first 10 classes.

The visualization reveals that LaCLIP trained with our proposed LLM-based rewriting strategy exhibits clearer class boundaries and more distinct clusters compared to other approaches. This observation suggests that language augmentations not only enhance the performance of text encoders, but also improve the ability of vision encoders to learn a more effective image embedding space that is well-suited for downstream tasks.

## G. Detailed Experiment Results for LaCLIP-MT

In Table A9, we present a detailed performance comparison among CLIP, LaCLIP, and the Multi-Text version LaCLIP-MT, as introduced in Section 5.

Table A10: Performance comparison of CLIP and LaCLIP trained with different backbone architectures, ViT-S/16, ViT-B/16 and ViT-L/16, on CC12M.

(a) Zero-shot and Linear-probing Experiment Results

| Backbone | Model | Food-101 | CIFAR-10 | CIFAR-100 | SUN397 | Cars | Aircraft | DTD | Pets | Caltech-101 | Flowers | STL-10 | EuroSAT | RESISC45 | GTSRB | Country211 | Average | ImageNet |
|---|---|---|---|---|---|---|---|---|---|---|---|---|---|---|---|---|---|---|
| | | | | | | | | Zero-shot | | | | | | | | | | |
| ViT-S/16 | CLIP | 44.0 | 54.7 | 32.6 | 41.9 | 20.2 | 2.5 | 20.1 | 56.9 | 74.0 | 29.3 | 88.0 | 29.1 | 36.0 | 11.2 | 4.4 | 36.3 | 36.9 |
| | LaCLIP | **57.6** | **70.6** | **37.1** | **55.6** | **29.1** | **6.6** | **29.7** | **71.2** | **81.1** | **39.5** | **93.6** | 26.7 | **40.0** | **14.7** | **8.4** | **44.1** | **46.3** |
| ViT-B/16 | CLIP | 50.8 | 64.9 | 38.5 | 44.7 | 24.1 | 2.4 | 19.4 | 64.1 | 77.4 | 33.2 | 91.0 | 20.1 | 38.9 | 7.3 | 5.1 | 38.8 | 40.2 |
| | LaCLIP | **60.7** | **75.1** | **43.9** | **57.0** | **36.3** | **5.6** | **31.0** | **72.4** | **83.3** | **39.9** | **95.1** | **27.3** | **44.3** | **12.7** | **8.9** | **46.2** | **48.4** |
| ViT-L/16 | CLIP | 54.1 | 76.0 | 44.3 | 49.7 | 31.2 | 3.4 | 20.9 | 65.8 | 79.9 | 34.7 | 92.6 | **30.6** | 41.1 | 9.0 | 6.1 | 42.6 | 44.0 |
| | LaCLIP | **60.5** | **80.4** | **47.3** | **58.1** | **38.8** | **5.7** | **31.0** | **71.5** | **82.0** | **39.6** | **95.8** | 18.6 | **46.8** | **13.0** | **9.2** | **46.6** | **49.1** |
| | | | | | | | | Linear-Probing | | | | | | | | | | |
| ViT-S/16 | CLIP | 78.9 | 91.7 | 75.3 | 70.5 | 69.1 | 46.5 | **74.4** | 84.3 | 90.8 | 94.8 | 96.3 | **95.9** | **91.7** | 76.5 | 17.9 | 77.0 | 67.1 |
| | LaCLIP | **80.3** | **93.0** | **76.6** | **71.8** | **73.0** | **49.0** | 74.3 | **85.3** | **91.8** | **95.1** | **97.0** | 95.4 | 90.7 | **78.4** | **18.2** | **78.0** | **69.1** |
| ViT-B/16 | CLIP | 81.6 | 93.8 | 79.3 | 72.0 | 75.1 | 52.6 | 75.6 | 86.2 | 92.2 | 95.3 | 97.3 | **96.7** | **93.1** | 80.6 | 19.7 | 79.4 | 70.3 |
| | LaCLIP | **82.9** | **94.7** | **79.7** | **73.8** | **79.9** | **54.5** | **75.7** | **87.7** | **93.0** | **96.4** | **98.0** | 96.4 | 93.0 | **81.9** | 19.7 | **80.5** | **72.3** |
| ViT-L/16 | CLIP | 83.5 | 95.3 | 81.4 | 73.4 | 80.1 | 57.8 | 76.8 | 88.4 | 93.3 | 96.5 | 97.9 | 97.0 | **94.0** | 82.9 | **20.8** | 81.3 | 72.9 |
| | LaCLIP | **83.8** | **95.8** | **82.8** | **74.4** | **81.4** | **58.1** | **77.2** | **88.6** | **93.9** | **97.2** | **98.2** | 97.0 | 93.7 | **85.2** | 20.5 | **81.9** | **73.7** |

(b) Few-shot Experiment Results

| Backbone | Model | Food-101 | CIFAR-10 | CIFAR-100 | SUN397 | Cars | Aircraft | DTD | Pets | Caltech-101 | Flowers | STL-10 | EuroSAT | RESISC45 | GTSRB | Country211 |
|---|---|---|---|---|---|---|---|---|---|---|---|---|---|---|---|
| ViT-S/16 | CLIP | 85.4±0.6 | 75.1±0.6 | 81.4±0.7 | 97.1±0.3 | 89.2±0.5 | 58.5±1.0 | 83.2±0.6 | 91.0±0.5 | 97.6±0.3 | 97.5±0.2 | 91.9±0.4 | 82.9±0.5 | 90.7±0.5 | 67.9±0.8 | 43.5±0.7 |
| | LaCLIP | **88.3±0.5** | **79.4±0.6** | **81.7±0.7** | **97.7±0.2** | **94.0±0.4** | **65.2±1.0** | **84.5±0.6** | **92.4±0.5** | **98.4±0.2** | **98.0±0.2** | **95.5±0.3** | **81.7±0.5** | **91.3±0.4** | **72.2±0.8** | **46.5±0.7** |
| ViT-B/16 | CLIP | 87.0±0.5 | 77.5±0.6 | 82.1±0.7 | 97.2±0.2 | 90.9±0.5 | 62.0±1.0 | 83.3±0.6 | 91.1±0.5 | 98.2±0.2 | 97.6±0.2 | 92.6±0.4 | **83.4±0.5** | 91.2±0.4 | 70.6±0.8 | 44.3±0.7 |
| | LaCLIP | **89.9±0.5** | **81.3±0.5** | **85.0±0.6** | **98.0±0.2** | **95.3±0.3** | **68.1±1.0** | **84.9±0.6** | **93.4±0.4** | **98.9±0.2** | **98.4±0.2** | **95.9±0.2** | 83.0±0.5 | **92.4±0.4** | **76.4±0.8** | **46.7±0.7** |
| ViT-L/16 | CLIP | 89.1±0.5 | 81.1±0.5 | 84.8±0.6 | 97.8±0.2 | 93.0±0.5 | 66.4±1.0 | 84.3±0.6 | 93.2±0.4 | 98.7±0.2 | 98.2±0.2 | 93.4±0.3 | 84.6±0.5 | 92.2±0.4 | 74.1±0.8 | 45.2±0.7 |
| | LaCLIP | **90.3±0.4** | **84.5±0.5** | **86.4±0.6** | **98.0±0.2** | **95.6±0.3** | **70.5±1.0** | **84.6±0.6** | **94.6±0.4** | **99.1±0.1** | **98.8±0.2** | **96.0±0.2** | **85.0±0.5** | **92.8±0.4** | **78.9±0.8** | **47.2±0.7** |

The pre-training was performed on CC12M and RedCaps datasets. The results highlight the potential of the multi-text version of the CLIP loss to enhance the performance of LaCLIP even further. By pairing each image with all corresponding texts, the vision encoder receives more diverse supervision during training iterations. he improvements are particularly significant for the RedCaps dataset, where LaCLIP-MT achieves an additional 1.9% increase in zero-shot classification accuracy on ImageNet.

## H. Detailed Experiment Results for Different Backbone

In Table A10, we present the detailed experiment results on CC12M using different backbone architectures, including ViT-S/16, ViT-B/16, and ViT-L/16 encoders. The results consistently demonstrate that our proposed LaCLIP outperforms the vanilla CLIP baseline across all backbone architectures. This highlights the scalability of LaCLIP, as it consistently improves performance on various downstream tasks while leveraging encoders of different sizes.

## I. Ablation on LLaMA model

We performed two ablation studies on the LLaMA model to assess the impact of modifying key components on the performance of LaCLIP. The studies focused on two factors: model size and temperature. By systematically investigating these factors, we aimed to shed light on their influence and provide valuable insights into the effectiveness and adaptability of the LLM-based augmentation approach. All experiments were conducted on LaCLIP using a single text augmentation strategy

Table A11: Ablation study on LaCLIP trained with text rewrites generated with different LLaMA model size on CC12M.

(a) Zero-shot and Linear-probing Experiment Results

| Model Size | Food-101 | CIFAR-10 | CIFAR-100 | SUN397 | Cars | Aircraft | DTD | Pets | Caltech-101 | Flowers | STL-10 | EuroSAT | RESISC45 | GTSRB | Country211 | Average | ImageNet |
|---|---|---|---|---|---|---|---|---|---|---|---|---|---|---|---|---|---|
| | | | | | | | Zero-shot | | | | | | | | | | |
| N/A (CLIP) | 50.8 | 64.9 | 38.5 | 44.7 | 24.1 | 2.4 | 19.4 | 64.1 | 77.4 | 33.2 | 91.0 | 20.1 | 38.9 | 7.3 | 5.1 | 38.8 | 40.2 |
| 7B | 57.0 | 71.1 | 38.9 | 51.2 | **31.6** | 3.9 | 25.5 | 63.0 | 80.8 | **36.9** | 92.9 | 24.5 | 39.6 | 10.1 | 6.9 | 42.3 | 44.5 |
| 13B | 55.4 | 71.5 | **39.3** | 51.3 | 29.6 | 4.0 | **26.4** | **65.7** | 80.7 | 36.0 | 93.8 | 17.0 | 38.7 | 9.0 | **7.6** | 41.7 | **44.8** |
| 33B | 56.7 | **76.0** | 37.7 | **52.0** | 31.2 | **4.5** | 24.3 | 60.7 | **80.9** | 35.4 | **94.4** | 26.7 | **40.4** | 11.6 | 7.0 | 42.6 | 44.4 |
| 65B | **57.5** | 69.2 | 38.9 | 51.6 | 31.1 | 4.1 | 25.3 | 65.2 | 79.0 | 36.8 | 93.1 | **31.7** | 40.2 | **15.0** | 7.4 | **43.1** | 44.4 |
| | | | | | | | Linear-Probing | | | | | | | | | | |
| N/A (CLIP) | 81.6 | 93.8 | 79.3 | 72.0 | 75.1 | 52.6 | 75.6 | 86.2 | 92.2 | 95.3 | 97.3 | 96.7 | 93.1 | 80.6 | 19.7 | 79.4 | 70.3 |
| 7B | 81.8 | **94.3** | 79.7 | 73.3 | 77.5 | 55.0 | 75.4 | 87.4 | 92.5 | **96.3** | **97.6** | 96.9 | 92.6 | 81.3 | **20.2** | 80.1 | 71.2 |
| 13B | 82.1 | 93.7 | 78.2 | 73.0 | 77.6 | **55.6** | 74.6 | 87.4 | **92.7** | 96.0 | 97.4 | 96.3 | **93.2** | 82.5 | 20.0 | 80.0 | 71.2 |
| 33B | 81.8 | 94.1 | 79.4 | 73.3 | 78.6 | 54.1 | 75.0 | 86.4 | 92.4 | 96.1 | 97.3 | 96.6 | 93.1 | 81.5 | 19.8 | 80.0 | **71.4** |
| 65B | **82.2** | 94.2 | 79.3 | 73.0 | **78.7** | 54.0 | 75.4 | 87.3 | 91.9 | 95.4 | 97.5 | 96.7 | 92.7 | **82.5** | 20.0 | 80.1 | 71.3 |

(b) Few-shot Experiment Results

| Model Size | Food-101 | CIFAR-10 | CIFAR-100 | SUN397 | Cars | Aircraft | DTD | Pets | Caltech-101 | Flowers | STL-10 | EuroSAT | RESISC45 | GTSRB | Country211 |
|---|---|---|---|---|---|---|---|---|---|---|---|---|---|---|---|
| N/A (CLIP) | $87.0_{\pm0.5}$ | $77.5_{\pm0.6}$ | $82.1_{\pm0.7}$ | $97.2_{\pm0.2}$ | $90.9_{\pm0.5}$ | $62.0_{\pm1.0}$ | $83.3_{\pm0.6}$ | $91.1_{\pm0.5}$ | $98.2_{\pm0.2}$ | $97.6_{\pm0.2}$ | $92.6_{\pm0.4}$ | $83.4_{\pm0.5}$ | $91.2_{\pm0.4}$ | $70.6_{\pm0.8}$ | $44.3_{\pm0.7}$ |
| 7B | $88.8_{\pm0.5}$ | $78.4_{\pm0.6}$ | $83.3_{\pm0.6}$ | $97.7_{\pm0.2}$ | $93.4_{\pm0.4}$ | $66.5_{\pm1.0}$ | $84.4_{\pm0.6}$ | $92.5_{\pm0.4}$ | $98.6_{\pm0.2}$ | $98.0_{\pm0.2}$ | $94.3_{\pm0.3}$ | $84.0_{\pm0.5}$ | $\mathbf{92.3_{\pm0.4}}$ | $\mathbf{73.7_{\pm0.8}}$ | $45.6_{\pm0.7}$ |
| 13B | $\mathbf{89.1_{\pm0.5}}$ | $79.2_{\pm0.6}$ | $82.8_{\pm0.7}$ | $\mathbf{97.9_{\pm0.2}}$ | $94.0_{\pm0.4}$ | $66.3_{\pm1.0}$ | $84.1_{\pm0.6}$ | $92.9_{\pm0.4}$ | $98.5_{\pm0.2}$ | $\mathbf{98.2_{\pm0.2}}$ | $94.4_{\pm0.3}$ | $83.2_{\pm0.5}$ | $91.6_{\pm0.4}$ | $73.6_{\pm0.8}$ | $45.7_{\pm0.7}$ |
| 33B | $88.6_{\pm0.5}$ | $\mathbf{80.3_{\pm0.6}}$ | $\mathbf{83.6_{\pm0.6}}$ | $97.8_{\pm0.2}$ | $\mathbf{94.3_{\pm0.4}}$ | $65.4_{\pm1.0}$ | $\mathbf{84.7_{\pm0.6}}$ | $92.8_{\pm0.4}$ | $98.6_{\pm0.2}$ | $\mathbf{98.2_{\pm0.2}}$ | $\mathbf{94.5_{\pm0.3}}$ | $84.2_{\pm0.5}$ | $92.1_{\pm0.4}$ | $72.0_{\pm0.8}$ | $\mathbf{45.8_{\pm0.7}}$ |
| 65B | $88.8_{\pm0.5}$ | $79.2_{\pm0.6}$ | $82.9_{\pm0.6}$ | $97.8_{\pm0.2}$ | $94.1_{\pm0.4}$ | $\mathbf{66.6_{\pm1.0}}$ | $84.3_{\pm0.6}$ | $\mathbf{93.1_{\pm0.4}}$ | $98.6_{\pm0.2}$ | $98.1_{\pm0.2}$ | $\mathbf{94.5_{\pm0.3}}$ | $\mathbf{85.6_{\pm0.5}}$ | $91.9_{\pm0.4}$ | $72.5_{\pm0.8}$ | $45.6_{\pm0.7}$ |

with the *ChatGPT* meta-input-output prompting pairs. The models were pre-trained on the CC12M dataset.

**Model Size.** Given that LLaMA offers multiple models with varying numbers of parameters, including 7B, 13B, 33B, and 65B, it is widely acknowledged that larger models tend to excel in NLP tasks involving reasoning and comprehension. Building upon this observation, we sought to explore the potential benefits of incorporating larger LLaMA models into our framework, with the aim of enhancing the performance of LaCLIP on downstream tasks.

To investigate whether the use of larger LLaMA models would yield improved results, we conducted a series of experiments where LaCLIP was trained using text augmented by LLaMA models of different sizes. We compared the performance of LaCLIP across these different configurations and summarized the results in Table A11.

Through our analysis, we have observed that even the smallest and relatively lightweight LLaMA model (7B) is sufficient to significantly boost the performance of LaCLIP on CLIP. Although larger LLaMA models showed some improvement on certain downstream datasets, the overall impact was relatively modest in our experimental setups focused on training vision-language models. It is worth mentioning that different model sizes may benefit from different temperature settings during the sampling process, and we leave this as a topic for future research. In the following sections, we specifically examine the effect of temperature on the 7B model.

**Temperature.** The temperature parameter plays a crucial role in the LLaMA token sampling process as it controls the balance between diversity and precision in the generated text. Higher values of temperature increase text diversity, but excessively high values can introduce random words or non-English tokens, negatively impacting the results.

Table A12: Ablation study on LaCLIP trained with text rewrites generated with different LLaMA temperature on CC12M.

(a) Zero-shot and Linear-probing Experiment Results

| temperature | Food-101 | CIFAR-10 | CIFAR-100 | SUN397 | Cars | Aircraft | DTD | Pets | Caltech-101 | Flowers | STL-10 | EuroSAT | RESISC45 | GTSRB | Country211 | Average | ImageNet |
|---|---|---|---|---|---|---|---|---|---|---|---|---|---|---|---|---|---|
| Zero-shot | | | | | | | | | | | | | | | | | |
| 0.3 | 52.1 | 66.6 | **40.1** | 44.0 | 30.0 | 4.0 | 22.3 | 62.2 | 79.7 | 34.8 | 90.7 | 21.3 | 37.3 | 10.9 | 6.3 | 40.2 | 43.6 |
| 0.5 | 54.0 | 69.5 | 36.4 | 46.1 | **31.8** | 3.4 | 22.9 | 62.3 | 80.2 | 35.8 | 93.0 | 22.4 | 38.1 | 10.9 | 6.1 | 40.9 | 44.0 |
| 0.7 | 53.6 | 67.2 | 37.5 | 48.3 | 31.5 | 3.9 | 24.0 | 63.5 | 78.6 | 34.6 | 91.9 | 24.2 | **42.9** | 8.1 | 6.7 | 41.1 | 43.8 |
| 0.9 | **57.0** | 71.1 | 38.9 | 51.2 | 31.6 | 3.9 | **25.5** | 63.0 | **80.8** | **36.9** | 92.9 | **24.5** | 39.6 | 10.1 | 6.9 | **42.3** | **44.5** |
| 1.1 | 55.8 | **72.8** | 39.2 | **53.1** | 28.6 | **4.2** | 23.6 | **64.7** | 80.6 | 34.2 | **93.1** | 21.8 | 37.4 | **15.2** | **7.6** | 42.1 | 44.0 |
| Linear-Probing | | | | | | | | | | | | | | | | | |
| 0.3 | **82.1** | 94.0 | 79.0 | 72.9 | 77.9 | 54.9 | 75.3 | **87.6** | **92.7** | 96.2 | 97.5 | 96.7 | 92.8 | **81.9** | 19.6 | **80.1** | 71.1 |
| 0.5 | **82.1** | 94.0 | 79.2 | 72.6 | 78.3 | 53.7 | **75.7** | 86.8 | 92.0 | 95.9 | 97.5 | 96.6 | **93.2** | 81.5 | 19.7 | 79.9 | 71.0 |
| 0.7 | 81.9 | **94.3** | 78.9 | 73.2 | **78.7** | 54.7 | 75.6 | 86.8 | 92.4 | 96.0 | 97.5 | 96.5 | 92.8 | 80.6 | 19.9 | 80.0 | 71.2 |
| 0.9 | 81.8 | **94.3** | **79.7** | 73.3 | 77.5 | **55.0** | 75.4 | 87.4 | 92.5 | **96.3** | **97.6** | **96.9** | 92.6 | 81.3 | **20.2** | **80.1** | 71.2 |
| 1.1 | 81.7 | 94.0 | 78.8 | **73.4** | 77.2 | 54.0 | 74.3 | 87.0 | 92.2 | 95.7 | **97.6** | 96.1 | 93.1 | 80.4 | 20.1 | 79.7 | **71.3** |

(b) Few-shot Experiment Results

| temperature | Food-101 | CIFAR-10 | CIFAR-100 | SUN397 | Cars | Aircraft | DTD | Pets | Caltech-101 | Flowers | STL-10 | EuroSAT | RESISC45 | GTSRB | Country211 |
|---|---|---|---|---|---|---|---|---|---|---|---|---|---|---|---|
| 0.3 | $89.1_{\pm0.5}$ | $77.8_{\pm0.6}$ | $82.3_{\pm0.7}$ | $97.6_{\pm0.2}$ | $93.3_{\pm0.4}$ | $66.1_{\pm1.0}$ | $84.3_{\pm0.6}$ | $\mathbf{93.0}_{\pm0.4}$ | $98.5_{\pm0.2}$ | $98.1_{\pm0.2}$ | $93.6_{\pm0.3}$ | $83.8_{\pm0.5}$ | $91.8_{\pm0.4}$ | $71.9_{\pm0.8}$ | $45.3_{\pm0.7}$ |
| 0.5 | $88.6_{\pm0.5}$ | $77.8_{\pm0.6}$ | $82.4_{\pm0.7}$ | $97.6_{\pm0.2}$ | $93.2_{\pm0.4}$ | $65.5_{\pm1.0}$ | $84.1_{\pm0.6}$ | $92.9_{\pm0.4}$ | $98.5_{\pm0.2}$ | $98.1_{\pm0.2}$ | $93.8_{\pm0.3}$ | $85.0_{\pm0.5}$ | $92.0_{\pm0.4}$ | $71.9_{\pm0.8}$ | $45.4_{\pm0.7}$ |
| 0.7 | $88.8_{\pm0.5}$ | $78.0_{\pm0.6}$ | $82.2_{\pm0.7}$ | $\mathbf{97.8}_{\pm0.2}$ | $93.3_{\pm0.4}$ | $65.5_{\pm1.0}$ | $84.1_{\pm0.6}$ | $92.5_{\pm0.5}$ | $98.6_{\pm0.2}$ | $98.1_{\pm0.2}$ | $93.9_{\pm0.3}$ | $84.0_{\pm0.5}$ | $91.8_{\pm0.4}$ | $72.5_{\pm0.8}$ | $\mathbf{45.7}_{\pm0.7}$ |
| 0.9 | $\mathbf{88.8}_{\pm0.5}$ | $\mathbf{78.4}_{\pm0.6}$ | $\mathbf{83.3}_{\pm0.6}$ | $97.7_{\pm0.2}$ | $\mathbf{93.4}_{\pm0.4}$ | $\mathbf{66.5}_{\pm1.0}$ | $\mathbf{84.4}_{\pm0.6}$ | $92.5_{\pm0.4}$ | $98.6_{\pm0.2}$ | $98.0_{\pm0.2}$ | $94.3_{\pm0.3}$ | $\mathbf{84.0}_{\pm0.5}$ | $\mathbf{92.3}_{\pm0.4}$ | $\mathbf{73.7}_{\pm0.8}$ | $45.6_{\pm0.7}$ |
| 1.1 | $88.7_{\pm0.5}$ | $80.1_{\pm0.5}$ | $83.6_{\pm0.7}$ | $\mathbf{97.8}_{\pm0.2}$ | $\mathbf{93.4}_{\pm0.4}$ | $64.8_{\pm1.0}$ | $83.8_{\pm0.6}$ | $92.5_{\pm0.4}$ | $\mathbf{98.7}_{\pm0.2}$ | $98.1_{\pm0.2}$ | $\mathbf{95.2}_{\pm0.3}$ | $82.3_{\pm0.5}$ | $91.4_{\pm0.4}$ | $70.7_{\pm0.8}$ | $\mathbf{45.7}_{\pm0.7}$ |

We conducted experiments with temperature values ranging from 0.3 to 1.1, and the detailed results of employing different temperatures for LLaMA generation are provided in Table A12. The results show that overall the performance is quite robust across temperatures. Generally as the temperature increases, the performance initially improves, reaching a peak around $\tau = 0.9$, and then begins to decline. Therefore, $\tau = 0.9$ appears to be the optimal temperature for text rewriting in the context of text augmentation, and we consistently use this value in all of our experiments.

## J. Ablation on Non-contrastive Training

Language rewrites techniques used in LaCLIP holds potential for broader applications. In order to evaluate the impact of the language augmentation strategy on non-contrastive vision-language pre-training methods, we integrated this strategy into Virtex's training pipeline [14], leading to the formation of Language-augmented Virtex (La-Virtex). We replicated the identical setup in their official implementation and trained the two models on the CC12M dataset. Table A13 shows the linear classification performance on the PASCAL VOC07 dataset of the two pre-trained models. The result demonstrates that the incorporation of language rewrites in La-Virtex outperforms the standard Virtex. This indicates that language augmentation could potentially be more generic and beneficial to non-contrastive vision-language model training methods as well.

Table A13: Comparison of Virtex training with and without Language Augmentation on VOC07 classification.

| Model | Language Aug | VOC07 |
|---|---|---|
| Virtex | ✗ | 78.40 |
| La-Virtex | ✔ | **80.92** |

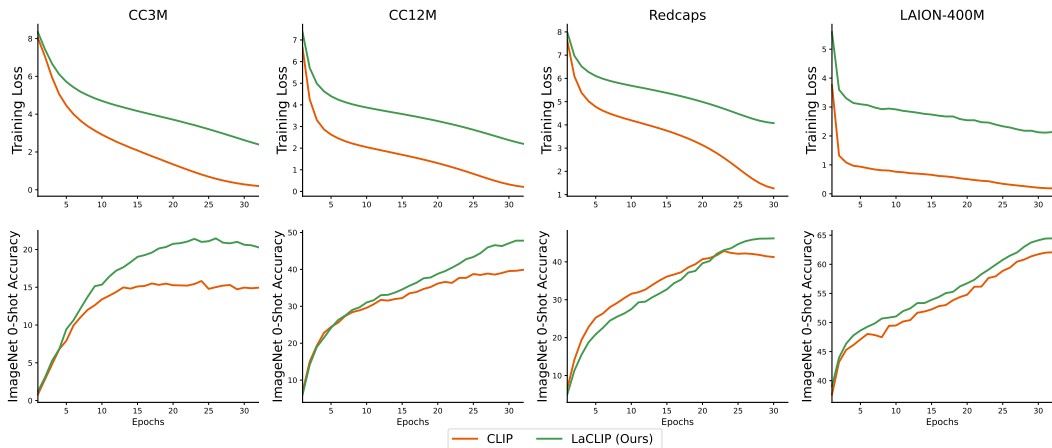

Figure A2: Training loss and validation accuracy for CLIP and LaCLIP trained on various datasets including CC3M, CC12M, RedCaps and LAION-400M. Top row is the training loss curve, bottom row is the validation accuracy, measured by zero-shot accuracy on ImageNet. Each column corresponds to a specific training dataset. For each figure, X-axis is the training epoch.

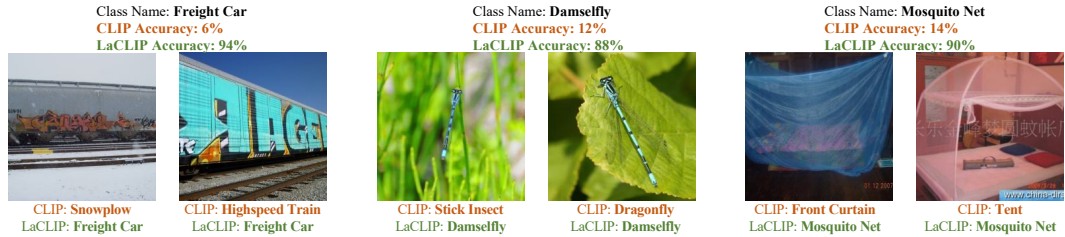

Figure A3: Visualization of image examples corrected by LaCLIP on the ImageNet validation set.

## K. Training and Validation Curves

We show the training and validation curve on CC3M, CC12M, RedCaps and LAION-400M datasets in Figure A2. The results demonstrate LaCLIP consistently achieves higher validation accuracy and higher training loss across different datasets. This indicates language augmentation is improving the model's generalization ability rather than its optimization process. This is because Language augmentations used in LaCLIP makes the pre-training task more challenging and therefore improves the generalization ability of the model.

## L. Visualization of Examples being Corrected

To provide a qualitative understanding of the categories most impacted by language augmentation, we present examples from the three categories with the most significant accuracy improvements by LaCLIP on the ImageNet validation set in Figure A3. This demonstrate LaCLIP has the ability to distinguish between some fine-grained categories where vanilla CLIP faces challenges.

## M. Detailed Experiment Results for Pre-trained Text-encoder

In Table A14, we present the detailed experiment results on CC12M between LaCLIP and CLIP models trained with different text encoder setups. For the CLIP models where pre-trained text encoders are used, we replaced the text encoder and tokenizer with the pre-trained BERT-Base model while keeping all other parameters to be the same. The experiments with pre-trained BERT encoder are conducted in two distinct setups: (a) fine-tuning the entire model, and (b) freezing the weights of the text encoder. The results shows using the pre-trained text encoder as initialization and fine-tune

Table A14: Ablation study between LaCLIP and CLIP models trained with different pre-trained Text Encoder (**Text-Enc**) setups on CC12M. For the models with a pre-trained text encoder, we use the pre-trained BERT-base model.

(a) Zero-shot and Linear-probing Experiment Results

| Method | Text-Enc Pre-trained | Text-Enc Frozen | Food-101 | CIFAR-10 | CIFAR-100 | SUN397 | Cars | Aircraft | DTD | Pets | Caltech-101 | Flowers | STL-10 | EuroSAT | RESISC45 | GTSRB | Country211 | Average | ImageNet |
|---|---|---|---|---|---|---|---|---|---|---|---|---|---|---|---|---|---|---|---|
| | | | | | | | | Zero-shot | | | | | | | | | | | |
| CLIP | ✗ | ✗ | 50.8 | 64.9 | 38.5 | 44.7 | 24.1 | 2.4 | 19.4 | 64.1 | 77.4 | 33.2 | 91.0 | 20.1 | 38.9 | 7.3 | 5.1 | 38.8 | 40.2 |
| CLIP | ✔ | ✗ | 55.3 | 66.8 | 38.1 | 50.3 | 33.1 | 4.2 | 25.9 | 64.8 | 78.6 | 34.5 | 92.5 | **27.9** | 39.2 | **13.5** | 7.0 | 42.1 | 42.9 |
| CLIP | ✔ | ✔ | 15.0 | 57.0 | 23.9 | 30.8 | 5.1 | 2.6 | 17.6 | 16.4 | 54.8 | 5.2 | 85.4 | 19.5 | 27.4 | 3.0 | 4.0 | 24.5 | 23.2 |
| LaCLIP | ✗ | ✗ | **60.7** | **75.1** | **43.9** | **57.0** | **36.3** | **5.6** | **31.0** | **72.4** | **83.3** | **39.9** | **95.1** | 27.3 | **44.3** | 12.7 | **8.9** | **46.2** | **48.4** |
| | | | | | | | | Linear-Probing | | | | | | | | | | | |
| CLIP | ✗ | ✗ | 81.6 | 93.8 | 79.3 | 72.0 | 74.9 | 52.7 | 75.6 | 86.2 | 92.2 | 95.3 | 97.3 | **96.7** | 93.1 | 80.5 | 19.7 | 79.4 | 70.3 |
| CLIP | ✔ | ✗ | 81.5 | 93.6 | 79.1 | 72.5 | 76.7 | 54.2 | 74.3 | 86.4 | 92.1 | 95.8 | 97.0 | 96.6 | **93.2** | 79.3 | **20.0** | 79.5 | 70.4 |
| CLIP | ✔ | ✔ | 74.9 | 91.9 | 74.4 | 68.9 | 60.0 | 48.0 | 71.0 | 79.6 | 87.5 | 90.7 | 96.2 | 95.6 | 91.1 | 75.7 | 18.1 | 74.9 | 66.0 |
| LaCLIP | ✗ | ✗ | **82.9** | **94.7** | **79.7** | **73.8** | **79.9** | **54.5** | **75.7** | **87.7** | **93.0** | **96.4** | **98.0** | 96.4 | 93.0 | **81.9** | 19.7 | **80.5** | **72.3** |

(b) Few-shot Experiment Results

| Method | Text-Enc Pre-trained | Text-Enc Frozen | Food-101 | CIFAR-10 | CIFAR-100 | SUN397 | Cars | Aircraft | DTD | Pets | Caltech-101 | Flowers | STL-10 | EuroSAT | RESISC45 | GTSRB | Country211 | Average |
|---|---|---|---|---|---|---|---|---|---|---|---|---|---|---|---|---|---|---|
| CLIP | ✗ | ✗ | 87.0 | 77.5 | 82.1 | 97.2 | 90.9 | 62.0 | 83.3 | 91.1 | 98.2 | 97.6 | 92.6 | **83.4** | 91.2 | 70.6 | 44.3 | 83.3 |
| CLIP | ✔ | ✗ | 87.8 | 75.5 | 81.6 | 97.5 | 91.7 | 64.9 | 83.5 | 91.9 | 98.3 | 97.6 | 93.3 | 83.0 | 91.6 | 71.5 | 44.6 | 83.6 |
| CLIP | ✔ | ✔ | 80.2 | 75.0 | 78.6 | 96.8 | 82.9 | 60.7 | 80.4 | 85.7 | 96.7 | 93.9 | 94.5 | 79.5 | 87.9 | 67.1 | 44.8 | 80.3 |
| LaCLIP | ✗ | ✗ | **89.9** | **81.3** | **85.0** | **98.0** | **95.3** | **68.1** | **84.9** | **93.4** | **98.9** | **98.4** | **95.9** | 83.0 | **92.4** | **76.4** | **46.7** | **85.8** |

the entire model can bring some benefit to vanilla CLIP training, while freezing the text encoder weights will result in performance drop. In the meantime, LaCLIP shows superior performance and outperforms all pre-trained text encoder counterparts, demonstrating the effectiveness of language augmentation strategy.

