In this paper, we propose LaCLIP, an approach that leverages LLMs to generate text rewrites and improve CLIP training through language augmentations. While the training process itself does not entail any additional memory or computation overhead compared to vanilla CLIP, the process of generating text rewrites using LLMs can be computationally expensive, requiring significant GPU resources and taking hours for large datasets. This could potentially contribute to a higher carbon footprint. Additionally, the quality of the rewritten text generated by LLaMA is not filtered, which may result in some irrelevant details that do not align well with the corresponding images. This misalignment could impact the transferability of the learned embeddings to downstream tasks.

To address these limitations, future work could focus on developing more efficient methods for generating text rewrites using LLMs, reducing the computational burden without sacrificing performance. Furthermore, techniques for filtering the rewritten texts could be explored, aiming to retain only the most relevant and accurate versions while discarding those with misleading details. This would enable the model to learn a better embedding space that is robust and transferable across different downstream datasets, improving overall performance and alignment between vision and text encoders.