# OpenReview forum: "Improving CLIP Training with Language Rewrites"
_NeurIPS.cc/2023/Conference — NeurIPS 2023 poster_

### Official Review · Reviewer_qFtF · 2023-07-04

**Soundness:** 4 excellent
**Presentation:** 4 excellent
**Contribution:** 3 good
**Rating:** 7
**Confidence:** 5

**Summary:**

CLIP uses data augmentation for image inputs but neglects the diversity of texts associated with the same image. To overcome this limitation, this paper introduces Language augmented CLIP (LaCLIP), a simple yet highly effective approach that enhances CLIP training through the startegy of language re-writing. Extensive experiments conducted on CC3M, CC12M, RedCaps, and LAION-400M datasets demonstrate the significance of LaCLIP, which outperforms CLIP by 8.2% on CC12M and 2.4% on LAION-400M in terms of ImageNet 0-shot acc. without imposing additional computation or memory overhead during training.

**Strengths:**

Overall, this is a good paper as it has notable contributions to the community. Vanilla CLIP often exhibits an imbalance where the image encoder is strong while the language encoder is comparatively weaker. It is partially because CLIP has very diverse image inputs yet lacks proper data augmentation strategies for language. This work presents a very interesting solution, i.e., language re-writing with a third-party language model or human efforts. The motivation behind the paper is commendable, as it is addressing a very important problem for vision-language pre-training. The method LaCLIP is simple and easy to follow. The experiments are extensive, and the results are very significant.

**Weaknesses:**

My major concern lies in whether LaCLIP could be regarded as a method of knowledge transfer. Specifically, while augmenting the captions by a large language model, the CLIP text encoder actually distills some knowledge from that. So, I am wondering if your LaCLIP outperforms vanilla CLIP simply because LaCLIP has a stronger text encoder. To ablate this problem, we can first load or distill from a pre-trained language model as CLIP's text encoder and then perform its training process.

There are also some minor drawbacks: LaCLIP relies on a well-pretrained language re-writer such as ChatGPT and LLaMA, which incurs heavy computing cost for language data augmentation and prevents end-to-end training.

**Questions:**

1. How is the difference between your LaCLIP and the paradigm mentioned in "Weaknesses"?
2. Does a stronger LLM always yield better LaCLIP performance?

**Limitations:**

See "Weaknesses".

---

> ### Author Rebuttal · Authors · 2023-08-10
>
> We thank the reviewer very much for their positive comments and insightful suggestions.
>
> **[Q1. Pre-trained Text Encoder]**
> Intuitively, leveraging pre-trained text encoder could be beneficial to CLIP training given its intrinsic understanding of textual context. However, earlier investigations into Locked-Image Tuning (LiT) [1] have conducted meticulous ablations, revealing that incorporating pre-trained text encoders, such as the BERT model in their case, may not yield substantial benefits.
> For comprehensive insights, we direct your attention to Figure 3 (top right) in their [paper](https://arxiv.org/pdf/2111.07991.pdf),
> The legend accompanying the figure aids in its interpretation, where each letter on the right corresponds to a distinct text encoder condition. To elaborate:
> * **U:** The Encoder is pre-trained
> * **u:** The Encoder is randomly initialized
> * **L:** The Encoder is pre-trained and its parameters are frozen
>
> It is crucial to focus on three distinctive lines in ImageNet 0-shot:
> * **uu (Orange dotted line):** Baseline performance of the vanilla CLIP .
> * **uU (Orange line):** Fine-tuning using a pre-trained text encoder.
> * **uL (Gray dotted line):** Pre-trained text encoder's parameters are frozen.
>
> Based on these results, fine-tuning based on pre-trained text models doesn'tnotably improve CLIP performance. Also, freezing the parameters of the pre-trained text model even leads to performance drop when compared to training from scratch.
>
> Contrarily, our experiments demonstrate that training LaCLIP with language rewrites induces far more substantial performance improvements. This highlights the need for explicit sentence augmentation instead of relying on implicit knowledge from pre-trained text models, thus showcasing the effectiveness of our LaCLIP approach.
>
> [1] Zhai, X., Wang, X., Mustafa, B., Steiner, A., Keysers, D., Kolesnikov, A. and Beyer, L., 2022. Lit: Zero-shot transfer with locked-image text tuning. In CVPR 2022.
>
> **[Q2.Heavy computing cost]**
> Acknowledging the potential computational expense of pre-computing language rewrites using LLMs is important. For instance, on a single machine equipped with 8 A100 GPUs, the rewriting of the CC3M dataset requires approximately 7 hours.
> However, it's crucial to note that this rewriting procedure is a one-time operation for each image-text dataset. Consequently, all CLIP models trained on the same dataset can subsequently reap the benefits without incurring additional computational overhead.
> To facilitate further research, we are planning to release all the precomputed rewritten texts for the datasets used in our paper. This proactive step eliminates the need for researchers to undergo the rewriting process again, streamlining their work and promoting the advancement of the field.
>
>
> **[Q3. Stronger LLM]**
> The potential strengths of LLMs can be attributed to two factors: **increased model size** and **better alignment**. We address the impact of larger model sizes in Appendix I, with summarized results in the table below:
>
> < Table S1>. Zero-Shot performance on LaCLIP trained with rewrites generated with different LLaMA model size on CC12M.
>
> | Model Size | Downstream | ImageNet |
> |:----------:|:----------:|:--------:|
> | N/A (CLIP) | 38.8       | 40.2     |
> | 7B         | 42.3       | 44.5     |
> | 13B        | 41.7       | 44.8     |
> | 33B        | 42.6       | 44.4     |
> | 65B        | 43.1       | 44.4     |
>
> We observed that even employing the smallest LLaMA model yields remarkable enhancements in vanilla CLIP training. While increasing the LLaMA model size can lead to performance gains on specific downstream tasks, the overall impact on larger model sizes remains relatively modest.
>
> Regarding improved alignment within LLMs, we believe that employing instruction-tuned models could introduce greater diversity into the rewritten text. We leave this exploration to future research endeavors.

---

> > ### Author Response · Authors · 2023-08-20
> > **Rebuttal follow up**
> >
> > Dear Reviewer qFtF,
> >
> > Thank you for acknowledging the contributions of our work as well as your thoughtful insights once again!
> >
> > In addition to our discussion in the rebuttal, we have done some additional experiments with regard to your previous suggestions on **Pre-trained Text Encoder**, we put our **new reults** here and hope you to find them interesting and convincing:
> >
> > We follow your suggestions and directly compare the pre-trained text encoder setting in our exact experiment setup on CC12M, to further show the difference of LaCLIP comparing to using pre-trained text encoders. We replaced the text encoder and tokenizer with pre-trained BERT-Base model, and kept all other parameters to be the same. We tested 2 setups: *fine-tuning* the whole model and *frozen* BERT weight:
> >
> > < Table S2>.  Zero-shot performance comparison between different pre-trained text encoder setups on CC12M
> >
> > | Method | Pre-trained Text Encoder | Text Encoder Freeze | Downstream | ImageNet |
> > |---|:--|:--|:-:|:-:|
> > | CLIP (Vanila) | N/A (from scratch) | No | 38.8 | 40.2 |
> > | CLIP (BERT-Fine-tune) | BERT-base | No | 42.1 | 42.9 |
> > | CLIP (BERT-Frozen) | BERT-base | Yes | 24.5 | 23.2 |
> > | LaCLIP | N/A (from scratch) | No | **46.2** | **48.4** |
> >
> > The observations align with the findings depicted in Figure 3 of LiT [1]. Fine-tuning the pre-trained BERT model exhibits some enhancement in CLIP training performance, whereas retaining the frozen BERT encoder substantially degrades performance. In contrast, LaCLIP consistently outperforms all BERT pre-training configurations, underscoring the necessity for explicit sentence augmentation strategies.
> >
> > [1] Zhai, X., Wang, X., Mustafa, B., Steiner, A., Keysers, D., Kolesnikov, A. and Beyer, L., 2022. Lit: Zero-shot transfer with locked-image text tuning. In CVPR 2022.
> >
> > With 1 day remaining in the discussion phase, we would like to confirm if our response successfully addressed your concerns. Furthermore, we would love to provide additional clearification or experiments that you'd like to see!
> >
> > We will integrate our rebuttal based on your suggestions into the final version of the paper, including the distinctions between LaCLIP and pre-trained text encoders, results with stronger LLMs, and discussion on the computational costs. The inclusion of these aspects is poised to significantly enhance the quailty of the paper!
> >
> > Your dedication of time and effort in reviewing our work is truly appreciated. Please let us know if you have any additional comments or questions!
> >
> > Best Wishes,
> >
> > Authors

---

> > > ### Comment · Reviewer_qFtF · 2023-08-20
> > > **Response to authors' rebuttal**
> > >
> > > Thank you for your rebuttal and new experiment results. My concerns are addressed. Again, I really like this paper, as it presents a very simple but insightful approach to improving VL models.

---

> > > > ### Author Response · Authors · 2023-08-21
> > > > **Glad to learn your concerns are addressed!**
> > > >
> > > > Dear Reviewer qFtF,
> > > >
> > > > We greatly appreciate your kind words and support for our work!
> > > > It brings us immense joy to hear that our responses have effectively addressed your concerns.
> > > >
> > > > Once more, we want to express our gratitude for both your valuable time and your helpful constructive feedback!
> > > >
> > > > Best,
> > > > Authors

---

### Official Review · Reviewer_FjR1 · 2023-07-08

**Soundness:** 4 excellent
**Presentation:** 4 excellent
**Contribution:** 3 good
**Rating:** 6
**Confidence:** 4

**Summary:**

This paper proposes a new language augmentation method for training CLIP. Specifically, a large language model is prompted, by a few examples, to generate rewrites for existing texts in image-text paired datasets. Gains are demonstrated on CC12M, RedCaps, LAION-400M, etc. compared with CLIP and SLIP.

**Strengths:**

1. The proposed solution looks simple, taking good advantage of existing pretrained LLMs.
2. Extensive training and evaluation experiments on multiple publicly available datasets and backbones, within a reasonable computation budget.
3. Good ablations on immediate variants of augmentation strategies and ICL strategies.


**Weaknesses:**

1. Missing comparison with or discussion on relations to existing text augmentation methods applied to CLIP training, e.g. DeCLIP [1] introduced self-supervision and multi-view supervision which generated 2x2 pairs with both image augmentation and text augmentation such as synonym replacement, random swap, and random deletion. LaViLa [2] explored augmenting with a rephraser and even re-captioning with vision conditioned large language models.
2. Text augmentation might be addressing the issue of limited text data in image-text paired datasets. How would text augmentation compare with finetuning or frozen text encoder pre-trained on large scale text data? e.g. with BERT, ROBERTA, or MPNET embeddings, instead of learning from scratch on image-text paired datasets.
3. The current method uses ICL on LLaMA, how about instruction tuned models, do we still need the three examples?
4. It is also interesting to see examples of how language augmentation helped the learning of visual features. What are the examples that were corrected?
5. Text encoder size worths ablating, since the proposed solution focuses on text augmentation.
6. Scaling is unconfirmed with a large dataset and a large backbone at the same time, but this is not doable given limited compute.

[1] Supervision Exists Everywhere: A Data Efficient Contrastive Language-Image Pre-training Paradigm. ICLR 2022.
[2] Learning Video Representations from Large Language Models. CVPR 2023.

**Questions:**

It is preferred if there are existing results addressing the questions in the weaknesses section, but there is no need to do extra training due to the computation needed.

There are related works on arxiv. Discussion is recommended but not required.
[1] A Fistful of Words: Learning Transferable Visual Models from Bag-of-Words Supervision. https://arxiv.org/abs/2112.13884.
[2] Improved baselines for vision-language pre-training. https://arxiv.org/abs/2305.08675.

**Limitations:**

Yes

---

> ### Author Rebuttal · Authors · 2023-08-10
>
> We thank the reviewer for their positive comments and insightful suggestions.
>
> **[Q1. Comparison with DeCLIP and LaViLa]**
> DeCLIP and LaViLa also explore text augmentation benefits, but their methodologies differ from ours. DeCLIP requires multiple encodings for each training image-text pair, resulting in increased training overhead. LaViLa incorporates a Narrator component that depends on video frames and necessitates additional training efforts.
>
> We made a fair comparison with them by training CLIP using text augmentations from DeCLIP or LaViLa. DeCLIP employs EDA [1] for text augmentations, and LaViLa utilizes an open-source T5-based paraphraser [2] as its rephraser. We provide the result in Table S1, where our LaCLIP consistently outperforms both baselines.
>
> < Table S1>. Zero-Shot comparison with text augmentation baselines
> | Augment       | Downstream | ImageNet |
> |---------------|:----------:|:--------:|
> | DeCLIP        | 40.6       | 41.2     |
> | LaViLa        | 40.4       | 41.9     |
> | LaCLIP (Ours) | **46.2**   | **48.4** |
>
> Additional comparison with EDA and backtranslation can be found in Table 4 of the paper and Appendix D. A comprehensive discussion of these aspects will be integrated into the final version.
>
> [1] Wei, J. and Zou, K., 2019. EDA: Easy Data Augmentation Techniques for Boosting Performance on Text Classification Tasks. In EMNLP 2019.
>
> [2] High-quality sentence paraphraser using transformers in nlp. https://huggingface.co/ramsrigouthamg/t5-large-paraphraser-diverse-high-quality
>
> **[Q2. Pretrained Text Encoder]**
> Intuitively, leveraging pre-trained text encoder could be beneficial to CLIP training given its intrinsic understanding of textual context. However, earlier investigations into Locked-Image Tuning (LiT) [3] have conducted meticulous ablations, revealing that incorporating pre-trained text encoders, such as the BERT model in their case, may not yield substantial benefits.
> For comprehensive insights, we direct your attention to Figure 3 (top right) in their [paper](https://arxiv.org/pdf/2111.07991.pdf),
> The legend accompanying the figure aids in its interpretation, where each letter on the right corresponds to a distinct text encoder condition. To elaborate:
> * **U:** The Encoder is pre-trained
> * **u:** The Encoder is randomly initialized
> * **L:** The Encoder is pre-trained and its parameters are frozen
>
> It is crucial to focus on three distinctive lines in ImageNet 0-shot:
> * **uu (Orange dotted line):** Baseline performance of the vanilla CLIP .
> * **uU (Orange line):** Fine-tuning using a pre-trained text encoder.
> * **uL (Gray dotted line):** Pre-trained text encoder's parameters are frozen.
>
> Based on these results, fine-tuning based on pre-trained text models doesn't notably improve CLIP performance. Also, freezing the parameters of the pre-trained text model even leads to performance drop when compared to training from scratch.
>
> Contrarily, our experiments demonstrate that training LaCLIP with language rewrites induces far more substantial performance improvements. This highlights the need for explicit sentence augmentation instead of relying on implicit knowledge from pre-trained text models, thus showcasing the effectiveness of our LaCLIP approach.
>
> [3] Zhai, X., Wang, X., Mustafa, B., Steiner, A., Keysers, D., Kolesnikov, A. and Beyer, L., 2022. Lit: Zero-shot transfer with locked-image text tuning. In CVPR 2022.
>
> **[Q3. Instruction tuned models]**
> For instruction tuned models we normally should not need three examples. ChatGPT, an exemplar of instruction-tuned models, can perform text rewriting without explicit examples. Nonetheless, it becomes unscalable when applied to datasets containing hundreds of millions of entries due to substantial financial and time costs incurred through API calls.
>
>
> **[Q4. Visualization of Examples being corrected]**
> We added visualization in the rebuttal Figure PDF.
>
>
> **[Q5. Ablation on Text Encoder Sizes]**
> Please refer to the rebuttal Figure PDF for experimental details. The results indicate that altering the text encoder alone lacks a significant impact on performance. Notably, for ImageNet zero-shot, an intriguing observation is that larger text encoders lead to a decline in CLIP's performance, while LaCLIP's performance improves. This suggests the potential for overfitting in Vanilla CLIP with larger text encoders, and LaCLIP could potentially mitigate this issue. However, the observed changes aren't significant enough to warrant advocating for text encoders larger than the Base model, considering the associated memory and computational overhead.
>
> **[Q6. Scaling with large dataset and large backbone]**
>
> Subsequent to the main paper deadline, we conducted experiments for LaCLIP with a ViT-B/16 backbone on the LAION-400M dataset:
>
> < Table S2>. Zeroshot performance of ViT-B/16 trained on LAION-400M.
> | Method | Downstream | ImageNet |
> |--------|:----------:|:--------:|
> | CLIP   | 65.4       | 67.0     |
> | LaCLIP | 68.5       | **69.4** |
>
> The result shows that LaCLIP improves CLIP training with larger backbones on the LAION-400M dataset. Notably, LaCLIP maintains a 2.4% performance gain with a bigger backbone.
>
> Our aspiration is to continually advance the frontier of state-of-the-art open-sourced CLIP models. We remain committed to exploring larger datasets and models trained using the LaCLIP framework.
>
> **[Q7. Discussion with more related works]**
>
> We thank the reviewer for noting relevant works.
> For the Bag-of-Words paper, its reliance on word-level operations limits diversity, while its scalability on larger datasets is unverified.
> Conversely, LLM-based augmentation leverages large language models to enhance content with comprehensive details.
> In the Improved baseline paper, multiple augmentations on the same image increase training costs. Also, their text augmentations focus on word-level operations, limit the potential for variance. A comprehensive discussion will be included in the Related Works section.

---

> > ### Author Response · Authors · 2023-08-20
> > **Rebuttal follow up**
> >
> > Dear Reviewer FjR1,
> >
> > Thank you for the positive feedback and the insightful suggestions again!
> >
> > We have conduct some **additional experiments** to further resolve your concerns with regard to **pre-trained text encoder**. Here is a summary of our new results:
> >
> > In addition to our discussion in the rebuttal, we directly compare the pre-trained text encoder setting in our exact experiment setup on CC12M, to further show the advantage of LaCLIP comparing to using pre-trained text encoders. We replaced the text encoder and tokenizer with pre-trained BERT-Base model, and kept all other parameters to be the same. We tested both of the 2 suggested setups: *fine-tuning* the whole model and *frozen* BERT weight:
> >
> > < Table S3>.  Zero-shot performance comparison between different pre-trained text encoder setups on CC12M
> >
> > | Method | Pre-trained Text Encoder | Text Encoder Freeze | Downstream | ImageNet |
> > |---|:--|:--|:-:|:-:|
> > | CLIP (Vanila) | N/A (from scratch) | No | 38.8 | 40.2 |
> > | CLIP (BERT-Fine-tune) | BERT-base | No | 42.1 | 42.9 |
> > | CLIP (BERT-Frozen) | BERT-base | Yes | 24.5 | 23.2 |
> > | LaCLIP | N/A (from scratch) | No | **46.2** | **48.4** |
> >
> > The observations align with the findings depicted in Figure 3 of LiT [1]. Fine-tuning the pre-trained BERT model exhibits some enhancement in CLIP training performance, whereas retaining the frozen BERT encoder substantially degrades performance. In contrast, LaCLIP consistently outperforms all BERT pre-training configurations, underscoring the necessity for explicit sentence augmentation strategies.
> >
> > [1] Zhai, X., Wang, X., Mustafa, B., Steiner, A., Keysers, D., Kolesnikov, A. and Beyer, L., 2022. Lit: Zero-shot transfer with locked-image text tuning. In CVPR 2022.
> >
> > In our previous rebuttal we meticulously followed the detailed suggestions provided by you, and have added all of the requested experiments, discussions and visualizations to make our work even more comprehensive. We will add all detailed response into the final version of the paper.
> >
> > Since there is only 1 day left in the discussion period, we would like to kindly ensure that the reviewer has seen our response, and eager to know whether there is additional clarification or experiments that the reviewer would like us to offer. We would be extremely grateful if the reviewer could consider favorably updating the review if our response effectively addressed your concerns.
> >
> > Thanks again for the effort and time you have dedicated to our work! Please let us know if you have additional comments or questions.
> >
> > Best Wishes,
> >
> > Authors

---

### Official Review · Reviewer_YRKL · 2023-07-10

**Soundness:** 4 excellent
**Presentation:** 4 excellent
**Contribution:** 3 good
**Rating:** 8
**Confidence:** 5

**Summary:**

This paper introduces a simple text augmentation strategy to train vision-language models.
Given an image-caption dataset, the core idea is to use an off-the-shelf LLM to "rewrite" image captions.
Since LLM outputs do not necessarily read like image captions, authors use in-context learning --
input a few example caption rewrites to the LLM in-context to generate the rewrite.
The authors use this augmentation to train CLIP, which they call "Language Augmented CLIP" (or LaCLIP).
LaCLIP significantly outperforms CLIP on multiple downstream tasks at various training data/model sizes.

**Strengths:**

I think this paper matches the quality of a typical publication at the NeurIPS conference.
I recommend acceptance; it is relevant to the conference audience and will spur exciting discussion in the community.
Below I highlight the main strengths of the paper:

1. **Simplicity:**
The proposed method is conceptually simple and empirically powerful.
It improves the performance of contrastive image-text models like CLIP and SLIP on many downstream vision tasks.

2. **Proposed approach has no 'online' training overhead:**
Language rewrites can be performed once for the entire training dataset 'offline' and saved to disk.
During training, the only extra overhead is loading all candidate captions per image (a few extra lines of text),
and doing a random number generator to select one candidate caption.
This overhead is negligible — considering this, the empirical improvements are very appealing.

3. **Experimental thoroughness:**
This paper is an excellent example of an empirically thorough study.
The authors are interested in answering a single general question: How much can text augmentation benefits current CLIP-style models?
The authors do an excellent exploration to answer this question:

    - Many design choices for language rewrites are experimented with, even a baseline that sources meta-prompts from human annotators!
    - This augmentation is plugged in with two approaches (CLIP and SLIP) to show its "drop-in" benefits.
    - Datasets and model architectures of different sizes are used to showcase the scaling behavior of this approach.
    - Empirical evaluations cover a variety of downstream classification tasks (zero-shot, linear probing, few-shot) and 15+ datasets.

4. **Excellent clarity in writing and presentation:**
All technical details for empirical analysis are well-stated and easy to follow.
The main paper and supplementary material have adequate implementation details to aid reproducibility.
All result tables are neatly organized to highlight the central messages to the reader efficiently.

**Weaknesses:**

I have a few questions/concerns which I believe should be addressed or acknowledged,
and other suggestions (optional) to improve the paper.

1. **Writing can be adjusted to decouple the approach from CLIP:**
The proposed approach is at its core, a data augmentation strategy for methods that train with image-text pairs.
This broadly encompasses methods that perform generative training, like image captioning (e.g. VirTex, BLIP) and masked language modeling (e.g. ICMLM).
The writing can be slightly reworded to convey this, along with mentioning that in this paper only one instantiation is considered -- contrastive models like CLIP.

2. **Connection with prior works like BLIP?**
BLIP and other such works perform iterative training through language rewrites, using a captioning model to _replace_ training captions.
The proposed approach uses language rewrites as text augmentations -- I believe that making this connection in the paper is useful.

3. **Proposed approach lacks a mechanism to remove noisy captions:**
This approach expands the set of candidate captions per image through language rewrites.
However, the original caption in training data may be noisy and uninformative, which is common in web-scale datasets.
This approach does very little to remove such captions, and may not generate a semantically relevant caption through rewrites
because the rewrites do not condition on the input image.

**Questions:**

I have one question related to the weaknesses I mentioned:
Have the authors tried training non-contrastive vision-language models with this training strategy?

**Limitations:**

The authors have discussed the limitations and broader impact of their method in the appendix.
I agree with the authors' assessment and believe that the discussion is sufficient.
However, I urge the authors to move it to the main paper and instead transfer any side experiments or implementation details in the appendix.

---

> ### Author Rebuttal · Authors · 2023-08-10
>
> We thank the reviewer very much for their positive comments and insightful suggestions.
>
> **[Q1. Improved Writing]**
> We extend our sincere appreciation to the reviewer for acknowledging the broader potential utility of our proposed approach. We also share the belief that the text augmentation strategy holds potential for significantly wider applications. In response to the suggestion, we will revise the main paper to present the approach in a more encompassing manner.
>
> **[Q2. Connection with BLIP]**
> We express our gratitude to the reviewer for establishing the connections with BLIP. Notably, the BLIP model family incorporates iterative image captioning within its training pipeline, which inevitably brings more computation and memory overhead during training. However, a key advantage of BLIP lies in the fact that the generated captions are intricately linked with the image content, potentially yielding more relevance than strategies centered solely on language augmentation. A comprehensive discussion elucidating these aspects will be thoughtfully integrated into the main paper.
>
> **[Q3. Remove noisy captions]**
> This is an insightful observation. Indeed, the challenge of noisy data is one that all CLIP-based methods could potentially encounter. However, the empirical findings suggest that, given the current dataset scale, the shared information tends to outweigh the noise present in the text. This dynamic enables the model to learn meaningful image-text embeddings.
>
> Intuitively, in the context of LaCLIP, if the original text contains noise, the subsequent rewritten text could exhibit reduced significance and relevance to the associated image. To explore this aspect, we conducted experiments to train a CLIP model on CC12M dataset, using only one version of the rewritten text. Notably, the rewritten text was kept constant throughout the training process, without applying text augmentation, to ensure a fair comparison.
>
> < Table S1>. Zero-shot performance of CLIP trained with real and rewritten captions
>
> | Caption   | Downstream | ImageNet |
> |:---------:|:----------:|:--------:|
> | Real      | 38.8       | 40.2     |
> | Rewritten | 39.0       | 40.9     |
>
> Surprisingly, the empirical results demonstrate that training CLIP models solely with rewritten text can yield comparable or even slightly superior performance compared to using real captions. This outcome underscores the advantages of text rewriting, where the inclusion of additional details within the rewrites outweighs the potential impact of noise present in the captions.
>
> We anticipate that future research could delve into methods for integrating image information into augmented texts during the rewriting process. Additionally, employing image-guided models to filter out noisy captions from datasets could be explored as a means of data cleaning.
>
>
> **[Q4. Non-contrastive training]**
> In alignment with the reviewer's suggestion, we have incorporated the Language Augmentation strategy into the training pipeline of Virtex [1], resulting in the formation of Language-augmented Virtex (La-Virtex). Due to the constrained timeframe for the rebuttal, we meticulously replicated the identical setup in their official implementation and proceeded to train two models on the CC12M dataset. Subsequently, we evaluated the performance of these models on PASCAL VOC07 through linear classification.
>
> < Table S2>. Comparison of Virtex training on VOC07 classification
>
> | Model     | VOC07 |
> |-----------|:-----:|
> | Virtex    | 78.40 |
> | La-Virtex | 80.92 |
>
> The data presented in Table S2 highlights that the incorporation of language rewrites surpasses the performance of vanilla Virtex. This outcome suggests that the language augmentation strategy could potentially be beneficial to non-contrastive vision-language model training pipelines as well.
>
> [1] Desai, K. and Johnson, J., 2021. Virtex: Learning visual representations from textual annotations. In CVPR 2021.

---

> > ### Author Response · Authors · 2023-08-20
> > **Rebuttal follow up**
> >
> > Dear Reviewer YRKL,
> >
> > Thank you so much for liking our work and sharing the insightful comments again!
> >
> > As we approach the final day of the discussion period, we are reaching out to you to ensure that our rebuttal effectively addressed your concerns. Moreover, we would love to provide further clarification or conducting additional experiments, should you deem it necessary.
> >
> >  We will follow your invaluable suggestions to revise our paper and include all of the additional disscusion and experiment insights from the rebuttal into the main paper. We will be meticulous in incorporating all discussions on limitations into the main paper as well. We firmly believe that the inclusion of these elements will undeniably fortify the strength of the paper.
> >
> > Once more, we extend our heartfelt gratitude for your dedication and the time you've graciously devoted to the review of our work! Please don't hesitate to let us know if there's any additional questions or suggestions!
> >
> > Best Wishes,
> >
> > Authors

---

### Official Review · Reviewer_Fifz · 2023-07-26

**Soundness:** 3 good
**Presentation:** 3 good
**Contribution:** 3 good
**Rating:** 6
**Confidence:** 4

**Summary:**

This paper introduces a straightforward yet highly effective language augmentation technique for the foundational vision-language pre-training model, CLIP. The authors focus on exploring the in-context learning capabilities of large language models, such as LLaMaA-7B, to produce four distinct rewrites for each text sample in the dataset. These rewrites are generated using ChatGPT, Bard, COCO, or Human meta-input-output text pairs, making the augmentation process relatively simple.

On the experimental front, the authors present compelling evidence of strong transfer performance across multiple benchmarks, which is unsurprising given the rationality of informative text data augmentation. However, the paper could be even more impactful if the authors provide more thorough calibrations as suggested in the weakness section.

Overall, this work contributes a valuable language augmentation method to the CLIP paradigm, and its potential recognition within the broader community could be further solidified by addressing the weaknesses and providing additional calibrations on why and how the proposed method work.

**Strengths:**

+ This work exhibits a clear and compelling motivation, along with a straightforward and elegant solution.

+ The experimental results presented in Table 1, Table 2, and Table 3 demonstrate the strength and efficacy of the proposed approach.

+ The detailed descriptions provided in the paper render it highly accessible and easy to follow for readers.

**Weaknesses:**

The authors of this paper primarily focus on reporting impressive results without conducting a thorough analysis of why their proposed method performs better. To enhance the clarity of their work and provide a deeper understanding of their approach, several key aspects need to be addressed.

- First, it is essential for the authors to include the training loss and validation loss curves in their analysis. This will help readers discern whether the proposed method improves optimization (by achieving smaller training and validation losses) or generalization (by achieving a smaller validation loss only).

- Second, the main difference between the source description and the rewritten description lies in enriching the presented concepts with more detailed descriptions. However, this approach also faces the risk of hallucination, potentially generating descriptions that do not accurately reflect the facts. The authors should thoroughly analyze this issue and propose methods to alleviate it, rather than relying solely on hand-crafted instructions to guide the large language models (LLMs).

- Furthermore, there is a glaring absence of critical ablation experiments in the paper. To provide a comprehensive evaluation of their method, the authors should consider the following groups of ablation experiments:

1. **The influence of the length of the rewritten description**: The authors could attempt to generate rewritten descriptions of varying lengths, such as 1.5x or 2x, or even up to 5x longer than the original descriptions. By reporting comparison results and analyzing the outcomes, readers can gain insights into the impact of description length on the model's performance.

2. **The influence of the text encoder model scale**: The authors currently opt for the smallest text encoder from CLIP, following previous work. However, it is vital to explore whether the rewritten descriptions may require a larger text encoder, especially for significantly longer rewrites. Providing detailed comparison results will offer valuable insights into the significance of the text encoder model's scale.

By addressing these critical points, the authors can strengthen their work's analysis, transparency, and overall contribution to the field.

**Questions:**

The authors should provide more analysis and discussion on why the proposed method performs better except for reporting straightforward comparison results.

**Limitations:**

Yes.

---

> ### Author Rebuttal · Authors · 2023-08-10
>
> We thank the reviewer for their positive comments and insightful suggestions. We have thoroughly addressed the concerns raised by the reviewer in the following. We hope that our responses will provide greater clarity about our work and contribute positively to the assessment.
>
> **[Q1. Whether LaCLIP improves optimization or generalization]**
> We added the training and validation curve in the rebuttal Figure PDF. LaCLIP achieves higher validation accuracy and higher training loss, indicating its focus on generalization rather than optimization. Language augmentation makes the task more challenging and therefore improves generalization. This concept will be elaborated in the main paper.
>
> **[Q2. The risk of hallucination and methods to alleviate it]**
> LaCLIP might face hallucination risk when instructing LLMs to add substantial details to the original caption. However, our experiments demonstrates that more rewrites consistently boost performance. This implies that the advantages of diversified details outweigh the potential counterfactual risk.
>
> However, we acknowledge the possibility of harm to model training as the number of augmentations increases significantly. To address this concern, we have devised a strategy to identify and filter out potential outliers.
>
> In this strategy, we first train a CLIP model on real image-text pairs. Then, we employ this model to assess rewritten texts. We identify texts with the lowest CLIP-score with the paired images, as these are more likely to contain harmful hallucinations affecting model performance.
>
> We proceeded to conduct an initial experiment where LaCLIP was solely trained using the selected augmented texts. This yielded an ImageNet Zero-shot performance of 43.7%. Interestingly, this model's performance is inferior to all configurations outlined in Table S1. This observation implies that while this exclusion approach might remove the least relevant rewritten texts, the diversity introduced by these rewrites seems to outweigh the potential negative impact of hallucinatory content.
>
> Another direction for future exploration is to develop language rewriting techniques that take into account the corresponding images. Furthermore, as LLMs continue to progress and enhance their ability to manage hallucinatory and counterfactual errors, we anticipate an improvement in the quality of rewritten texts. This advancement, in turn, could result in an overall performance boost for LaCLIP. We intend to incorporate this aspect of the discussion into the main body of the text to provide a more comprehensive understanding.
>
> Nevertheless, our proposed language augmentation strategy remains simple and highly effective, demonstrating consistent performance gains on large-scale datasets. Our recent experiments on the LAION-400M dataset with ViT-B/16 further highlight LaCLIP's ability to yield enhancements within the context of large models and extensive datasets. With confidence, we assert that our approach has the potential to push the limits of state-of-the-art CLIP models.
>
>
> **[Q3.1 Ablation Experiments on Length of Rewrites]**
> Given that our approach employs in-context learning on LLMs to generate text rewrites, controling the exact length of the outputs isn't a straightforward task. To delve into this aspect, we conducted an ablation study using the existing generated caption rewrites. The methodology involved ranking the four rewritten texts for each instance in the CC12M dataset according to their length. Subsequently, these texts were grouped into four categories, ranging from the shortest to the longest. We then proceeded to train LaCLIP using the original text along with each of these four groups of text, each having varying lengths. The length statistics (relative to the original texts) and the ensuing LaCLIP performance are outlined in Table S1.
>
> < Table S1>. LaCLIP Performance with different text rewrites length on CC12M
>
> | length       | 0.5x | 0.8x | 1.0x | 1.6x     |
> |:-------------|:----:|:----:|:----:|:--------:|
> | IN zero-shot | 44.4 | 44.5 | 45.0 | **45.2** |
>
> The results suggest longer rewrites tend to improve CLIP training more, and the benefit of the increased diversity of the augmented texts outweigh any potential drawbacks related to counterfactual hallucination.
>
> **[Q3.2 Ablation Experiments on Text Encoder scale]**
> > it is vital to explore whether the rewritten descriptions may require a larger text encoder
>
> Please refer to the rebuttal Figure PDF for experimental details. The results indicate that altering the text encoder alone lacks a significant impact on performance. Notably, for ImageNet zero-shot, an intriguing observation is that larger text encoders lead to a decline in CLIP's performance, while LaCLIP's performance improves. This suggests the potential for overfitting in Vanilla CLIP with larger text encoders, and LaCLIP could potentially mitigate this issue. However, the observed changes aren't significant enough to warrant advocating for text encoders larger than the Base model, considering the associated memory and computational overhead.
>
> > especially for significantly longer rewrites.
>
> We also study the effect of text encoder sizes on rewrite text lengths:
>
> < Table S2>. ImageNet zero-shot with different text encoders on different rewrite length
>
> | Text encoder \ Length | 0.5x | 0.8x | 1.0x | 1.5x |
> |-----------------------|:----:|:----:|:----:|:----:|
> | Base                  | 44.4 | 44.5 | 45.0 | 45.2 |
> | Large                 | 44.0 | 43.8 | 44.5 | 44.8 |
>
> The results indicate incorporating a single augmentation does not seem to be sufficient for mitigating the overfitting issue, and it's worth noting that employing larger text encoders is not recommended when relying solely on one augmentation.
>
> We fully concur that integrating these analyses into our LaCLIP evaluations will render the paper more comprehensive and insightful, thereby benefiting future research endeavors.

---

> > ### Comment · Reviewer_Fifz · 2023-08-13
> > **Good rebuttal**
> >
> > Thanks for your detailed responses. The author is encouraged to add the rebuttal contents to the main paper in the future.
> >
> > In light of the authors' response, I have adjusted the original rating from "Borderline Accept" to "Weak Accept."

---

> > > ### Author Response · Authors · 2023-08-14
> > > **Thank you!**
> > >
> > > We sincerely appreciate your positive feedback! We will surely add all the additional discussions and experiments into the next version of the paper.
> > >
> > > Thanks again for your insightful comments, please do not hesitate to let us know If there are any further clarifications or experiments we can offer.

---

> > > > ### Author Response · Authors · 2023-08-20
> > > > **Rebuttal follow up**
> > > >
> > > > Dear Reviewer Fifz,
> > > >
> > > > Thank you again for discussing with us!
> > > >
> > > > With one day remaining in the disscusion period, we would love to hear whether we can offer you any additional clearifications or experiments that you find interesting, or if you have any further suggestions to help us strength this work even more.
> > > >
> > > > Thanks again for the effort and time you have dedicated to our work and discussion! Please don't hesitate to let us know if you have any further questions.
> > > >
> > > > Best Wishes,
> > > >
> > > > Authors

---

### Author Rebuttal · Authors · 2023-08-10

We express our gratitude to all the reviewers for dedicating their time and for providing valuable feedback, positive comments, and insightful suggestions.

We are pleased to note that the reviewers recognized:

* The clear and compelling motivation behind our idea. [Fifz, qFtF]
* The simplicity and effectiveness of our proposed method. [YRKL, FjR1, qFtF]
* The comprehensive experiments that showcase the effectiveness of LaCLIP. [All reviewers]
* The manuscript's well-organized structure, clear writing, and reader-friendly presentation. [Fifz, YRKL]

We have included a rebuttal Figure PDF, including training and validation curve [Fifz], ablations on text encoder sizes [Fifz, FjR1], and visualizations on corrected samples on ImageNet validation set [FjR1].

We have also provided comprehensive responses to the specific queries raised by each reviewer. We hope our explanations can effectively address all the concerns raised by the reviewers. We extend our gratitude once again to all the reviewers for their valuable time and insightful feedback. Should there be any additional clarifications or experiments required, please don't hesitate to let us know. We sincerely hope that our efforts will result in a favorable reconsideration of the scores by the reviewers.

---

### Author Response · Authors · 2023-08-18
**Looking forward to follow-up discussion**

Dear Reviewers,

We would like to thank you again for your effort and insightful comments!

As the deadline for the discussion period is approaching, we would like to inquire if there are any additional clarifications or experiments that we can provide. We would greatly appreciate it if you could consider favorably updating the review if our response successfully addressed your concerns.

Once again, we extend our heartfelt appreciation to the reviewers for their time and insightful suggestions. All the additional discussions will be incorporated in the next version of the paper.

Best,
Authors

---

### Decision · Program_Chairs · 2023-09-21

**Decision:**

Accept (poster)

**Comment:**

This paper presents a straightforward yet remarkably effective approach to augment text data, thereby enhancing the training of CLIP through a language re-writing strategy. Reviewers commend this work for its clear and elegant solution, along with the meticulous experimentation presented in the paper and the subsequent rebuttal. The reviewers applaud the diligence displayed in conducting additional experiments and providing clarity in response to several pivotal concerns raised during the review process, including an examination of the "risk of hallucination" and training loss, additional ablation studies on factors such as the length of the rewritten description, text encoder scale, and pretrained text encoder, as well as a comparative discussion with existing methods. These concerns have all been thoughtfully addressed, fortifying the updated version for publication.

In light of these comprehensive improvements, all reviewers have expressed scores above the acceptance threshold, and AC concurs with their recommendation to accept this paper. AC strongly encourages the authors to incorporate these valuable insights and constructive comments from the reviewers into the final version of the manuscript.